# Solvothermal synthesis of polyvinyl pyrrolidone encapsulated, amine-functionalized copper ferrite and its use as a magnetic resonance imaging contrast agent

Fatemeh Heydari[1,2], Ágnes M. Ilosvai[3,4], Noémi Kovács[1,2], Domokos Máthé[1,2☯], Ferenc Kristály[5], Lajos Daróczi[6], Zoltán Kaleta[7,8], Béla Viskolcz[3,4], Miklós Nagy[3], László Vanyorek[3], László Forgách[1,2☯*], Krisztián Szigeti[1,2☯]

1 Department of Biophysics and Radiation Biology, Semmelweis University, Budapest, Hungary, 2 In Vivo Imaging Advanced Core Facility, Hungarian Center of Excellence for Molecular Medicine (HCEMM), Szeged, Hungary, 3 Institute of Chemistry, University of Miskolc, Miskolc-Egyetemváros, Miskolc, Hungary, 4 Higher Education and Industrial Cooperation Centre, University of Miskolc, Miskolc, Hungary, 5 Institute of Mineralogy and Geology, University of Miskolc, Miskolc-Egyetemváros, Hungary, 6 Department of Solid-State Physics, University of Debrecen, Debrecen, Hungary, 7 Pro-Research Laboratory, Progressio Engineering Bureau Ltd., Szekesfehervar, Hungary, 8 Institute of Organic Chemistry, Semmelweis University, Budapest, Hungary

☯ These authors contributed equally to this work.
* forgach.laszlo@semmelweis.hu

**Data Availability Statement:** All relevant data are within the paper and its Supporting Information files.

## Abstract

Copper ferrite nanoparticles for use as MRI contrast agents were synthesized using two different methods. A novel microwave-assisted (MW) solvothermal method was developed and compared to a conventional 12-hour synthesis (Reflux) as an eco-friendlier approach. This innovative synthesis method successfully produced nanoparticles with enhanced properties compared to traditional ferrite materials. The nanoparticles' morphological and magnetic properties were evaluated and tested in in-vivo MRI studies. The results revealed both similarities and differences between the amine-functionalized copper ferrite nanoparticles. FTIR, XRD, HRTEM, and VSM analyses indicated improved properties in the $CuFe_2O_4$-$NH_2$ MW particles, while AFM confirmed successful polymer encapsulation of the nanoparticles. For the $CuFe_2O_4$-$NH_2$ MW sample, 76.8 wt% copper ferrite and 23.2 wt% magnetite were detected, with crystallite sizes of 8 ± 2 nm and 13 ± 2 nm, respectively. In the $CuFe_2O_4$-$NH_2$ Refl. sample, in addition to these two magnetic phases, larger copper particles (31.6 wt%) were also formed. DLS analysis demonstrated that the $CuFe_2O_4$-$NH_2$ MW sample exhibited excellent colloidal stability, maintaining its size distribution in aqueous media for 3 hours without aggregation, unlike the $CuFe_2O_4$-$NH_2$ Refl. sample, which showed slight aggregation. The $CuFe_2O_4$-$NH_2$ MW sample displayed superparamagnetic behavior (Ms: 15 emu/g, Mr: 0 emu/g, Hc: 0 Oe), while the $CuFe_2O_4$-$NH_2$ Refl. sample exhibited ferromagnetic characteristics (Ms: 40 emu/g, Mr: 1.35 emu/g, Hc: 30 Oe). Both samples produced comparable results during in vitro MRI measurements, showing similar T2* relaxation and signal characteristics. Further in vivo studies demonstrated that both samples induced significant hypointense changes. The study provides valuable insights into the synthesis, properties, and

**Funding:** This research was supported by the National Research, Development, and Innovation Fund (Hungary), within the TKP2021-NVA-14 project. F.H. received funding withing the framework of the TKP2021-EGA23. This work was supported by The European Union's Horizon 2020 Research And Innovation Program, grant agreements No. 739593: HCEMM, supported by the EU Programme: H2020-EU.4.a. and No. 859890 SmartAge ITN. Part of the work was also supported in the cadre of the iMatrix consortium at H2020-JTC2022 ERA-NET-Neuron. This work was also partly funded by grants from the Hungarian National Research, Development, and Innovation Office (Thematic Excellence Program, TKP-BIOImaging, financed under the 2020-4.1.1-TKP2020 funding scheme, Investment to the Future 2020.1.16-Jöv˝o-2021-00013 and TKP2021-EGA-23).

potential applications of these materials, emphasizing the importance of eco-friendly methods and the optimization of ferrite-based MRI contrast agents.

## Introduction

Copper ferrite nanoparticles, a subset of the well-known spinel ferrites, have garnered significant attention in recent years due to their outstanding magnetic, electrical, and optical properties. These nanoparticles, typically represented by the formula $MFe_2O_4$, where M denotes a divalent metal ion, possess a distinctive structure that has major implications for their use in various fields [1–3]. This article provides a comprehensive examination of two types of copper ferrite nanoparticles, focusing on their elemental composition, the synthesis techniques used, and how these processes influence their intrinsic properties in the context of their innovative applications as biomedical contrast agents [4–6].

The structural complexity of copper ferrite nanoparticles is determined by the cation distribution, expressed as $[Cu_x^{2+}Fe^{1-x3+}]A[Cu^{1-x2+}Fe^{1+x3+}]BO_4$, with the parameter 'x' playing a critical role in defining the crystal symmetry. Depending on the synthesis method, this cation arrangement can lead to either tetragonal or cubic symmetry [7,8]. These structural subtleties are essential for understanding the unique characteristics of these materials. Various methods have been explored for synthesizing copper ferrite nanoparticles, each offering its own benefits and drawbacks. Techniques range from solid-state reactions to sol-gel processes, sonochemical methods, and hydrothermal techniques, all of which enable fine control over the nanoparticles' size and shape [9–14]. However, many of these methods are complex, expensive, require high temperatures, and involve toxic reagents, which can limit their scalability and environmental sustainability.

For instance, hydrothermal processing is commonly used to produce magnetic nanoparticles. In this technique, the reaction mixture is placed in a Teflon-lined steel autoclave and subjected to high pressure and temperature [15,16]. Hydrothermal synthesis offers several advantages, including being cost-effective, environmentally friendly, and a one-step process that allows easy control over particle size [17,18]. However, this method can be time-consuming, with samples often requiring 12 to 20 hours at 200°C to form [19].

Another popular method for synthesizing magnetic ferrite nanoparticles is the co-precipitation technique. This method is widely used because it can produce large quantities of high-purity nanoparticles with controlled sizes [20,21]. One drawback of co-precipitation is that the resulting particles often have low crystallinity, however this can only be enhanced through combustion [20]. Also, like hydrothermal synthesis, co-precipitation tends to be time-intensive [22]. In the search for more efficient and environmentally friendly approaches, thermal treatment methods offer a promising alternative. These methods are straightforward, cost-effective, and eco-friendly, avoiding the use of toxic reagents, and making them suitable for industrial applications. For instance, Dhiwahar et al. employed microwave (MW) synthesis to produce zinc-doped copper ferrite nanoparticles. In this method, samples were microwaved at 800 W for 15 minutes, followed by cleaning with ethanol and deionized water, and calcining at 500°C for 2 hours, resulting in nanoparticles with a pure phase [23]. İbrahim synthesized cobalt ferrite nanoparticles using a combined microwave and combustion method, applying a 5-minute microwave treatment at 800 W, followed by annealing at 1000°C. This process revealed that high-temperature annealing significantly impacts the structural, morphological, and magnetic properties of the particles [24].

This article delves into the critical impact of solvothermal synthesis methods (microwave-assisted or reflux heating) on the properties of copper ferrite nanoparticles, providing insights

that expand upon and refine the thermal treatment process, and potential uses in biomedical applications, as MRI contrast agents.

MRI has been used widely in both preclinical research and clinical diagnosis, especially for using non-ionizing radiation, exceptional soft-tissue contrast, and the non-invasive nature of its imaging technique. Traditionally, MRI benefits from contrast agents, predominantly gadolinium chelates, to enhance images [25,26]. However, the emergence of concerns surrounding the nephrotoxicity of gadolinium-based contrast media, particularly in patients with chronic renal failure, has been the main motivation to explore further into finding an alternative contrast material [27,28]. In recent times, with the help of nanoscience magnetic nanoparticles, which are finely tuned in terms of size, shape, composition, and surface modifications, stand out as candidates of remarkable potential for MRI contrast agents [29–32]. What sets these nanoparticles apart is their ability to influence both T1-weighted and T2-weighted MRI, illuminating or darkening the resulting images, depending on the imaging sequence used [33,34]. Furthermore, they enable susceptibility-weighted imaging, introducing new possibilities in tumour detection [35–37]. Additionally, certain nanocomposites even transcend the conventional boundaries, functioning as bifunctional $T1 - T2$ or $T2^*$ contrast agents, promising enhanced diagnostic capabilities [38–41].

The evolution of MRI, operating at magnetic field strengths ranging from 7 to 9.4 Tesla, providing numerous advantages—higher signal-to-noise ratios and enhanced spectral resolution, establishing better spatial and temporal resolutions than previously attained [42–44]. In this context, the demand for optimized contrast agents is evident, presenting a significant opportunity for advancing the field of MRI [45].

To fine-tune MRI contrast agents to specific applications, research has intently focused on the modification of key nanoparticulate properties [46,47]. These properties include size, shape, composition, and capping agents. The composition of metal ferrite nanoparticles, represented as $MFe_2O_4$ (where M can be Ni, Co, Fe, Mn), holds a significant key to their efficacy as contrast agents [48]. It has been observed that the composition of these metal ferrite nanoparticles plays a crucial role in dictating MRI performance, with magnetization being a central determinant of transverse relaxivity (r2)–a vital factor for T2-weighted MRI [49–51]. The choice of capping agents that envelop nanoparticle surfaces has also been explored extensively. Small, hydrophilic coating molecules, such as ascorbate, citrate, and glutathione, have emerged as promising options [52–54]. Further, two-stage synthesis methods have been employed to render these nanoparticles water-soluble while preserving capping agent integrity [55,56].

As highlighted in the discussion of various synthesis methods, considerable focus has been placed on different solvothermal processes, with microwave synthesis being a commonly employed technique. In this study, two solvothermal synthesis approaches have been compared for producing amine-functionalized copper ferrite: microwave irradiation and reflux boiling. Both methods utilized the same precursors and quantitative ratios. The resulting magnetic nanoparticles are analysed and compared in terms of size, morphology, magnetization properties, colloidal stability, and phase composition. Additionally, their performance as MRI contrast agents are evaluated. These findings allow the assessment and comparing the effectiveness of the two synthesis methods in generating copper ferrite.

## Materials & methods

### Materials

The following precursors were applied during the solvothermal synthesis: copper(II) nitrate trihydrate, $Cu(NO_3)_2 \cdot 3H_2O$, MW: 241.6 g/mol (Fluka GmbH, D-30926 Seelze, Germany) iron (III) nitrate nonahydrate, $Fe(NO_3)_3 \cdot 9H_2O$ (VWR International, Leuven, Belgium) and

ethylene glycol, $HOCH_2CH_2OH$, (VWR Int. Ltd., F-94126 Fontenay-sous-Bois, France), monoethanolamine, $NH_2CH_2OH$ (Merck KGaA, D-64271 Darmstadt, Germany) and sodium acetate, $CH_3COONa$ (ThermoFisher GmbH, D-76870 Kandel, Germany). Polyvinylpyrrolidone (PVP) K30 (average M.W. 50.000 g/mol) was used for stabilizing agent (Acros Organics LtD/Thermo Fisher Scientific, B-2440, Geel, Belgium).

### Synthesis of the amine-functionalized magnetic nanoparticles

Amine-functionalized magnetic nanoparticles were synthesized using a solvothermal method, utilizing two different synthesis techniques (Fig 1). In the course of our previous work, magnetic nanoparticles have been produced many times and it was found that ferrites can be efficiently produced starting from nitrate salts [57,58]. The nitrate salts used, as copper and iron precursors, $CuFe_2O_4-NH_2$ Refl. sample was synthetized by refluxing and heating for 12 h, which is the synthesis method "A" (Fig 1). In the first step, iron (III) nitrate nonahydrate (10 mmol) and copper (II) nitrate trihydrate (5 mmol), along with sodium acetate (75 mmol), were dissolved in 50 mL of ethylene glycol (EG). The solution was then heated to 100°C in a round-bottom flask under reflux with continuous stirring. After 30 min, 20 mL (330.7 mmol) of monoethanolamine was added. The solution was continuously mixed and refluxed for 12 h. After cooling, the supernatant was separated by centrifugation (4200 RCF for 10 min). The solid phase was washed several times with distilled water, and the magnetic ferrite was easily

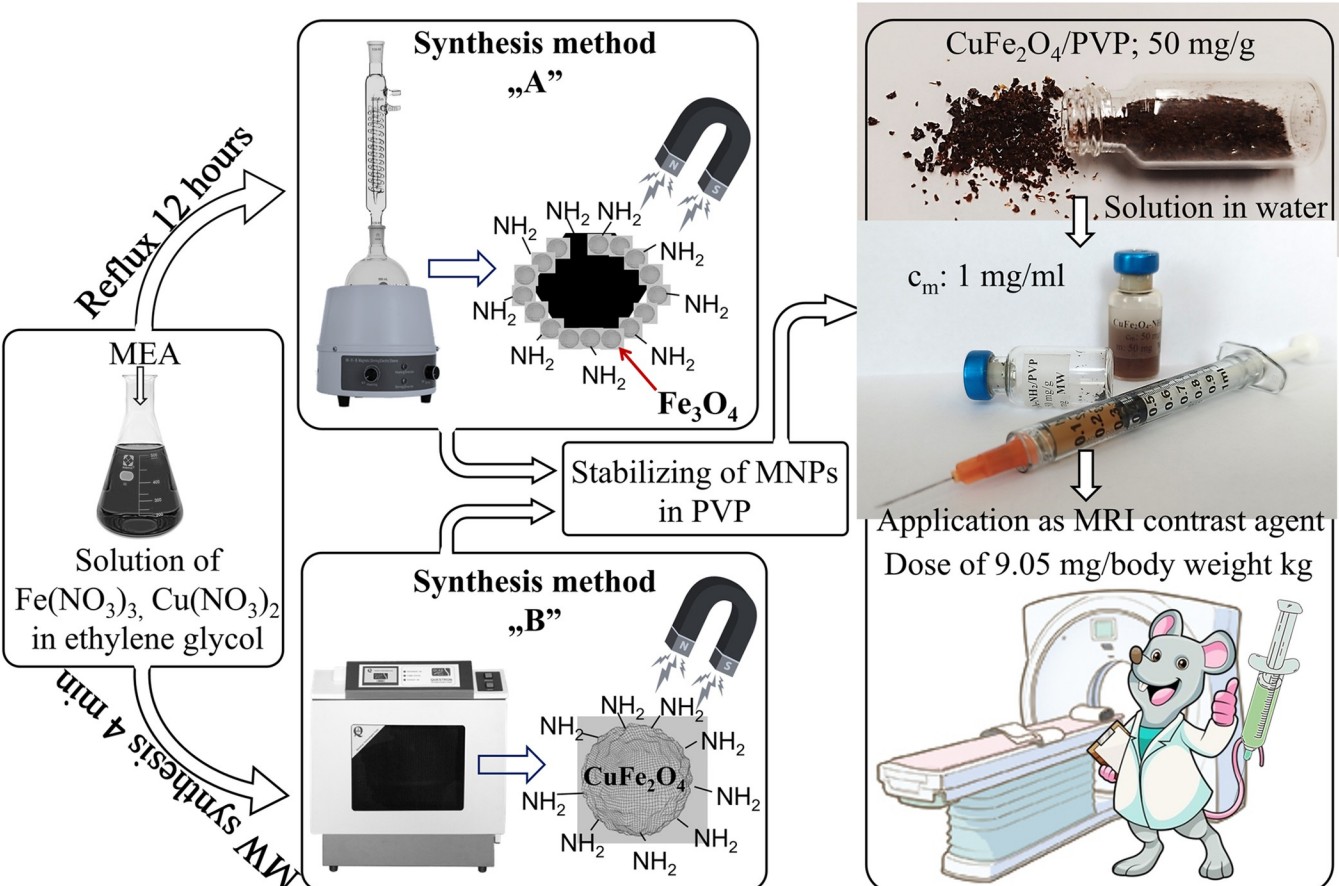

**Fig 1. Scheme of the preparation of NH₂-functionalized copper ferrite nanoparticles as MRI contrast agents.**

separated from the aqueous phase using a magnet. Finally, the ferrite sample was rinsed with 96 v/v% ethanol and dried overnight at 80°C. It was then suspended in distilled water to give copper ferrite concentration of 50 mg/mL.

The time consuming solvothermal method was modified to microwave assisted synthesis, thereby the 12 h synthesis time was reduced to only 4 min, that is the experimental scheme for synthesis method "B" shown in Fig 1. During the experiment, the same quantities of metal precursors, sodium acetate, EG, and monoethanolamine (MEA) were used. In Polytetrafluoroethylene (PTFE) digestion tubes (V = 50 mL) were carried the MW treatment of the reaction mixture, which were placed in a CEM MDS 81 D microwave digestion instrument. The MW synthesis was carried out at 200°C (350 W) for 4 min under atmospheric pressure. The $CuFe_2O_4$-$NH_2$ MW sample was separated and washed based on the above detailed method. From this, a 50 mg/mL copper ferrite dispersion in distilled water was prepared.

For steric stabilization, magnetic nanoparticles were embedded in polyvinylpyrrolidone. During this, 2.00 g of polyvinylpyrrolidone was dissolved in 50 mL of distilled water. After which 2 mL of the $CuFe_2O_4$-$NH_2$ dispersion (50 mg/mL) added. The PVP stabilized colloid was evaporated using a rotary vacuum evaporator at 70°C and 110 mbar. The ferrite contained PVP crystals were dried overnight at 80°C. The result was a reddish-brown, crystalline solid product that was highly soluble in water (Fig 1). And, the concentration of copper-ferrite nanoparticles stabilized in PVP support was 50 mg/g. Accordingly, 100 mg of PVP-stabilized copper ferrite contains 5 mg of $CuFe_2O_4$ nanoparticles. If 100 mg of PVP-supported copper ferrite sample was dispersed in 5 mL of distilled water (in injection vial), the concentration of the magnetic component is 1 mg/mL in stable colloid, which is applicable for MRI tests (Fig 1).

## Characterization techniques

Talos F200X G2 HRTEM (ThermoScientific, Waltham, MA, USA) with field emission electron gun (accelerating voltage: 20–200 kV) and SmartCam digital search camera (Ceta 16 Mpixel, 4k x 4k CMOS camera) and HAADF detector was applied for morphological characterization and size analysis. For the size analysis based on the TEM images, 100 particles were measured using a scalebar of pixel ratios from the TEM images. For the manual size determination, we used Image J softver [59–61]. The aqueous dispersion of the nanoparticles dropped onto 300 mesh copper grids (Ted Pella Inc., 4595 Redding, CA 96003, USA) during the sample preparation step. For identification and quantification of the crystalline phase of the samples, X-Ray Diffraction (XRD) was used with the help of a Bruker D8 equipment (Cu-Kα source) in parallel beam geometry (Göbel mirror) with Vantec detector (Bruker Corporation, Billerica, MA, USA). The average crystallite size of the domains was calculated by the mean column length calibrated method using the full width at half maximum (FWHM) and the width of the Lorentzian component of the fitted profiles. Identification of crystalline components were carried out by Search/Match, on ICDD PDF2 (2005) database in the Bruker DiffracPlus EVA software. Quantitative composition was calculated with Rietveld refinement in Bruker TOPAS4 by fundamental parameters approach (FPA) and the generated instrumental profile was validated on SRM640a Si standard. Crystallographic information for the identified phases was taken from ICSD and AMCSD databases [62–64].

Identification of the surface functional groups of the nanoparticles was executed with Fourier-Transform Infrared spectrometry (FTIR) by using Bruker Vertex 70 equipment. FTIR spectra were recorded in transmission mode in the 4000–400 wavenumber range at a resolution of 4 cm$^{-1}$ after 32 scans. For the FTIR analysis 10 mg sample was pelletized with 150 mg spectroscopic grade potassium bromide. The magnetic characterization of ferrite nanoparticles

was performed with a self-developed (University of Debrecen, Debrecen, Hungary) vibrating-sample magnetometer system based on a water-cooled Weiss-type electromagnet. The powder samples were pelletized for the measurements with typical mass of 20 mg. The magnetization (M) was measured as a function of magnetic field (H) up to 10000 Oe field strength at room temperature. The magnetization of paramagnetic particles can be described by the Langevin-function (Eq 1).

$$M(H) = M_s \left( \coth\left(\frac{m\mu_0 H}{kT}\right) - \frac{1}{\frac{m\mu_0 H}{kT}} \right)$$

Where: M magnetization of the material, Ms saturation magnetization, $Am^2/kg$ (emu/g) of the material, m magnetic moment, $Am^2$ (1 μB = $9.27401 * 10-24\ Am^2$) of an individual particle.

The imaging $CuFe_2O_4$-$NH_2$ crystallites using AFM was conducted using hundred-fold diluted samples on poly-L-lysine (PLL)-coated surfaces. The PLL-coated surface was prepared by dropping 100 μL of PLL (0.1% w/v) onto freshly cleaved mica. After an incubation of 20 min, repeated rinsing with purified water, and drying with a stream of high-purity nitrogen gas was performed. AFM images were collected in noncontact mode with a Cypher S instrument (Asylum Research, Santa Barbara, CA, USA) at 0.25–0.30 Hz line-scanning rate in air, using a silicon cantilever (OMCL AC-160TS, Olympus, Tokyo, Japan) oscillated at its resonance frequency (300–320 kHz). Temperature during the measurements was 25 ± 1˚C. AFM amplitude-contrast images are shown in this paper. The filter used on the images enhanced the details of the height amplitude contrast images (violet, orange, yellow, mud). AFM images were analyzed by using the built-in algorithms of the AFM driver software (Igor Pro, Wave Metrics Inc., Lake Oswego, OR, USA). Particle statistics was executed by analyzing a 2 μm × 2 μm height-contrast image with (n = 178) particles. Maximum height values were taken as the height of particles, and rectangularity was calculated as the ratio of the particle area to the area of a nonrotated inscribing rectangle. The closer a particle is to a rectangle, the closer this value is to unity.

DLS serves as a prevalent technique for quantifying the size distribution and colloidal stability of nanoparticles within a liquid suspension. By monitoring intensity fluctuations resulting from Brownian motion within the suspension, DLS elucidates crucial parameters: the hydrodynamic size (or apparent size), size distribution, and zeta potential. This method stands as a rapid means for quality assessment, offering insights into the colloidal properties of a liquid solution (67). The hydrodynamic diameter of the particles was determined using a Litesizer 500 (Anton Paar, Hamburg, Germany). DLS measurements were performed at 25˚C in automatic mode (for backscatter detector, fixed at 175˚; for side scatter, 90˚ detector angle; for front scatter, 15˚ detector angle) using a 633 nm He-Ne laser. Samples were measured in polystyrene disposable cuvettes (Anton Paar, Hamburg, Germany). During the measurements, the focus was kept steady. The measurement data were evaluated using software provided by the manufacturer, and statistical data and graphs were created and evaluated with Origin 9.0 (OriginLab) and Microsoft Excel 2013 software. DLS measurements were executed using 100 mg of samples, with the ferrite content in the PVP support of 50 mg/g. To these samples in vacuum vessels, 5 mL of ultrafiltered water was added to obtain a solution with a concentration of 1 mg/mL. This solution was diluted twenty-fold for the DLS measurements with ultrafiltered water.

Understanding tissue characteristics is based on measuring T2*, T2, and T1 relaxation times. T1 relaxation time, or spin-lattice relaxation time, refers to the time required for about 63% recovery of longitudinal magnetization post radiofrequency pulse disruption. It is distincted from T2 and T2* times, which are associated with transverse relaxation. T1 is

influenced by factors like water molecule mobility within tissues and proton density. The interplay between T1, T2*, and T2 times is vital for comprehensive tissue characterization in quantitative MRI studies. T1 contrast provides key data on tissue composition and integrity and is often used in therapeutic imaging due to variable T1 times across tissues. Conversely, T2 times, which can alter due to pathological conditions like tumors, are essential for diagnostic imaging. T1 emphasizes longitudinal magnetization recovery, while T2* and T2 times focus on transverse relaxation and are impacted by magnetic field inhomogeneities. Quantitative MRI is a powerful tool in both research and clinical practice because it can provide a comprehensive view of tissue properties by integrating data from T1, T2*, and T2 relaxation times. This approach helps researchers and clinicians gain valuable insights into various biological and pathological processes [65,66].

MRI measurements were performed with a nanoScan® PET/MR system (Mediso, Budapest, Hungary), having a 3 T magnetic field, 600 mT/m gradient system, and a volume transmit/receive coil with a diameter of 72 mm (for in vitro measurements) and 42 mm (for in vivo measurements). In vitro scans were performed on four different ferrite concentrations (0.02, 0.05, 0.1, and 0.2 mg/mL) of both samples ($CuFe_2O_4$-$NH_2$ Refl. and $CuFe_2O_4$-$NH_2$ MW) in 2 mL Eppendorf tubes. All relaxometry measurements were performed with the same geometrical parameters. One coronal slice was imaged with 4 mm of slice thickness, 50 mm of field of view, and 0.36 mm in-plane resolution. For determining T1 relaxation times, a Multi-IR FSE 2D sequence was used, with a repetition time of 5200 ms, echo time of 5.8 ms, and inversion times of 100, 400, 600, 800, 1000, 1300, and 2000 ms. The total measurement time was 21 min. The T2 relaxation times were determined using a Multi-echo SE 2D sequence with a repetition time of 3856 ms and a first echo time of 5.5 ms, followed by 31 echoes with echo spacing of 5.55 ms. The measurement time was 10 min. A Multi-echo GRE 2D sequence was used for the calculation of T2* relaxation times with a repetition time of 350 ms and a shortest echo time of 1.75 ms, which was followed by 31 echoes with echo spacing of 1.9 ms; the measuring time was 5 min.

For the in vivo measurements, n = 2 female, 10-week-old BalbC mice were used. The 0.2 mL of the 1 mg/mL contrast material was injected into the tail vein of the animals. In vivo measurements were performed with the mice under isoflurane anesthesia (5% for induction and 1.5–2% to maintain the appropriate level of anesthesia (Arrane®, Baxter, Newbury, UK). After the measurements, the animals were sacrificed using anaesthesia overdose of Euthasol.

The T2*-weighted gradient echo (GRE) scans were collected at two different time points (pre-injection and 10 min p.i.). A 80 × 50 mm FOV on 7 coronal slices was acquired, with a matrix size 256 × 200, slice thickness of 0.8 mm, slice gap of 0.2 mm, 4 averages and TR/ TE/ FA 270 ms/5.2 ms/90˚. The scan time was 3 min.

In InterView Fusion software (Mediso Ltd) 8 ROIs (liver, spleen, visceral fat tissue, renal pelvis, brain, muscle in the leg, cortex and medulla of the kidneys) were manually drawn on the pre-scans of each mouse. Images were normed to have same signal intensities in the muscle in order to have comparable signal intensity (SI) values. Relative signal changes were calculated in each ROI by the formula ($SI_{pre}$−$SI_{post}$) / $SI_{pre}$ to determine the biodistribution of the nanoparticles. The higher the measured signal intensity decrease, the more the accumulated nanoparticles.

The relaxation time maps (T1, T2, and T2*) and ROI based evaluation were performed in InterView Fusion (Mediso Ltd., Budapest, Hungary), and relaxivity calculations were done in Excel.

All procedures involving animals were conducted in accordance with the ARRIVE guidelines and the guidelines set by the European Communities Council Directive (86/609 EEC). The study was approved by the Animal Care and Use Committee of Semmelweis University

(PE/EA/01319-4/2023). Efforts were made to minimize animal suffering and to use the minimum number of animals necessary to achieve statistical significance.

The mice were anesthetized using isoflurane (5% for induction and 1.5–2% for maintenance) to ensure they remained unconscious and did not experience pain during the procedures. After the measurements, the animals were humanely sacrificed using an overdose of Euthasol, ensuring a painless and quick death.

## Results and discussion

### Characterization of the amine-functionalized magnetic nanoparticles

**Phase composition determination based on XRD measurements.** The possibilities of variations in phase composition may be due to the $CuFe_2O_4$-$NH_2$ Refl. and $CuFe_2O_4$-$NH_2$ MW samples were prepared using two different synthesis methods. To confirm that the samples consisted of nanoparticles with the desired magnetic properties, XRD measurements were conducted. XRD analysis is used for identifying potential contaminants, such as hematite or copper oxides, which are non-magnetic components that could compromise the quality of the contrast agent. Thus, the magnetic nano powder samples were examined by XRD for identification of the different crystalline phases (Fig 1A). In case of $CuFe_2O_4$-$NH_2$ Refl. sample reflection peaks were identified on the Rietveld refined diffractogram at 18.3˚ (111), 30.2˚ (220), 35.6˚ (311), 37.2˚ (222) 43.1˚ (400), 53.6˚ (422), 57.1˚ (511), and 62.7˚ (440) 2Θ degrees. This can be associated with the presence of the $CuFe_2O_4$ phase, which is the main component of the system, namely 35.9 wt% (Table 1). The average particle size of $CuFe_2O_4$-$NH_2$ Refl. sample was 18 ± 4 nm (PDF 77–0010). The above detailed reflexions, in the case of the magnetite phase were measured also by the help of Rietveld analysis (PDF 96–900–7645). The magnetite content of the samples was 32.6 wt%, where nanoparticles were 20 ± 5 nm. An additional set of reflexions were found on the diffractogram with high intensity at 43.3˚ and 50.5˚ 2Θ degrees, which belonged to the (111) and (200) Miller indexed Cu phase, with the amount of copper of 31.6 wt%. (PDF 04–0836). To be added, the XRD powder analysis yielded an average of 91 ± 21 nm crystallite size for Cu, peak shapes indicating a bimodal size distribution, which was revealed by TEM results, showing a minor fraction of greater (500–1100 nm) Cu crystals. The smaller size fraction was distributed between the ferrite crystals and/or on the edge of the larger euhedral crystals. In a previous publication, the preparation of zinc ferrite and its application as an MRI contrast agent was reported. The synthesis methods were very similar to those used in the present work for the preparation of copper ferrite. In contrast to the present results, no other phases were found in the sample for zinc ferrite, only the spinel was formed during both preparation methods [67]. In the case of copper(II) ions, reduction can occur, as ethylene glycol is an effective reducing agent in the conversion of Cu(II) ions to elemental copper nanoparticles. Ramyadewi et al. refluxed Cu(II) salt in ethylene glycol solution at 200˚C (as we did) and produced copper nanoparticles [68].

Summarizing the XRD results, it can be concluded that the nanoparticles produced by microwave synthesis do not contain elemental copper. They contain two magnetizable phases, magnetite and copper ferrite, both of which are well suited as MRI contrast agents. The refluxed sample contains copper particles in addition to the magnetic phases mentioned

**Table 1. Results of the size analysis (in nm) and quantity of each phase of the copper ferrite samples (based on XRD measurements).**

| Sample ID | $CuFe_2O_4$ d (nm) | wt% | $Fe_3O_4$ d (nm) | wt% | Cu d (nm) | wt% |
|---|---|---|---|---|---|---|
| $CuFe_2O_4$-$NH_2$ Refl. | 18 ± 4 | 35.9 | 20 ± 5 | 32.6 | 91 ± 21 | 31.6 |
| $CuFe_2O_4$-$NH_2$ MW | 8 ±2 | 76.8 | 13 ± 2 | 23.2 | - | - |

above. The magnetite has crystallized on the surface of the copper, as copper is a good thermal conductor and a magnetic layer has formed on its surface, this raises the possibility of testing it in magnetic hyperthermia for cancer treatment in the future.

On the diffractogram of the $CuFe_2O_4$-$NH_2$ MW sample, reflexions of the $CuFe_2O_4$ and the $Fe_3O_4$ phases were also identified (Fig 2B). The quantity of the cuprospinel was 76.8 wt%, and the average particle size was $8 \pm 2$ nm. The magnetite content of this sample was 23.2 wt%, and these nanoparticles were $13 \pm 2$ nm in size (Table 1). The magnetic nanoparticles produced by the two synthesis methods differ in size. In the case of $CuFe_2O_4$-$NH_2$ Refl. samples the synthesis time was 12 h at 200°C by which the crystal growth was facilitated, while the $CuFe_2O_4$-$NH_2$ MW sample was only 5 min exposed to the same conditions. Thus, larger crystals were developed by the longer exposure time.

## Morphological characterization and qualitative analysis of magnetic nanoparticles based on HRTEM measurements

The morphology and particle size of the amine-functionalized copper ferrite crystallites ($CuFe_2O_4$-$NH_2$ Refl.) were investigated by HRTEM (Fig 3). The HRTEM pictures clearly showed the spherical shape of the functionalized ferrite nanoparticles with $21 \pm 6$ nm average diameter based on the scalebar of the micrographs (Fig 3A and 3B). The average crystallite size of the previously mentioned ferrite crystallite was also calculated based on XRD measurements, which resulted in $18 \pm 4$ nm, similarly to the HRTEM results. The element maps were made from the ferrite nanoparticles, out of the position of each element (copper, iron and oxygen), which indicated the presence of copper ferrite nano crystallites (Fig 3D).

During electron microscopy examination, sample-areas with significant morphological differences in the $CuFe_2O_4$-$NH_2$ Refl. sample were identified. HRTEM pictures were showing large nano- and microplates found next to the above detailed copper ferrite nanoparticles. Based on the element analysis, these larger particles were identified as copper (Fig 4). The size of these Cu plates ranged between 500 nm and 1100 nm, as observed in the HRTEM pictures (Fig 4A–4C). The accumulation of small nanoparticles in the close proximity of the copper plates were detected. By applying the element analysis these small nanoparticles were identified as iron-oxide, namely magnetite nanoparticles, which was confirmed by XRD measurement (Fig 2A). The presence of copper (marked in green on the element map) was confirmed of the copper plate, but was not visible in its surroundings (Fig 4D). In contrast, signs of iron and oxygen were observed around the copper plates and on their surfaces. Based on these findings, it can be concluded that the copper particles were closely surrounded by magnetite nanoparticles. It is expected that the copper plates also respond to the magnetic field due to the presence of magnetite on their surface. The afore mentioned magnetite nanoparticles found in the close proximity of the copper plates were measured using the scale bar of the micrographs. The average particle size of $Fe_3O_4$ particles was determined to be $19 \pm 6$ nm, which is in compatible with the crystallite size obtained during XRD measurements ($20 \pm 5$ nm).

Based on the HRTEM pictures of the $CuFe_2O_4$-$NH_2$ MW samples, small nanoparticles were detected, but no copper pellets were found present on TEM picture of the $CuFe_2O_4$-$NH_2$ Refl. sample (Fig 5A–5C). On the element maps copper, iron and oxygen were present, which belonged to the copper ferrite and magnetite nanoparticles (Fig 5D).

## Identification of surface functional groups of nanoparticles based on FTIR measurements

The surface functional groups of the amine-functionalized ferrite and magnetite nanoparticles were identified using FTIR (Fig 6). In the FTIR spectra of the $CuFe_2O_4$-$NH_2$ Refl. sample, the

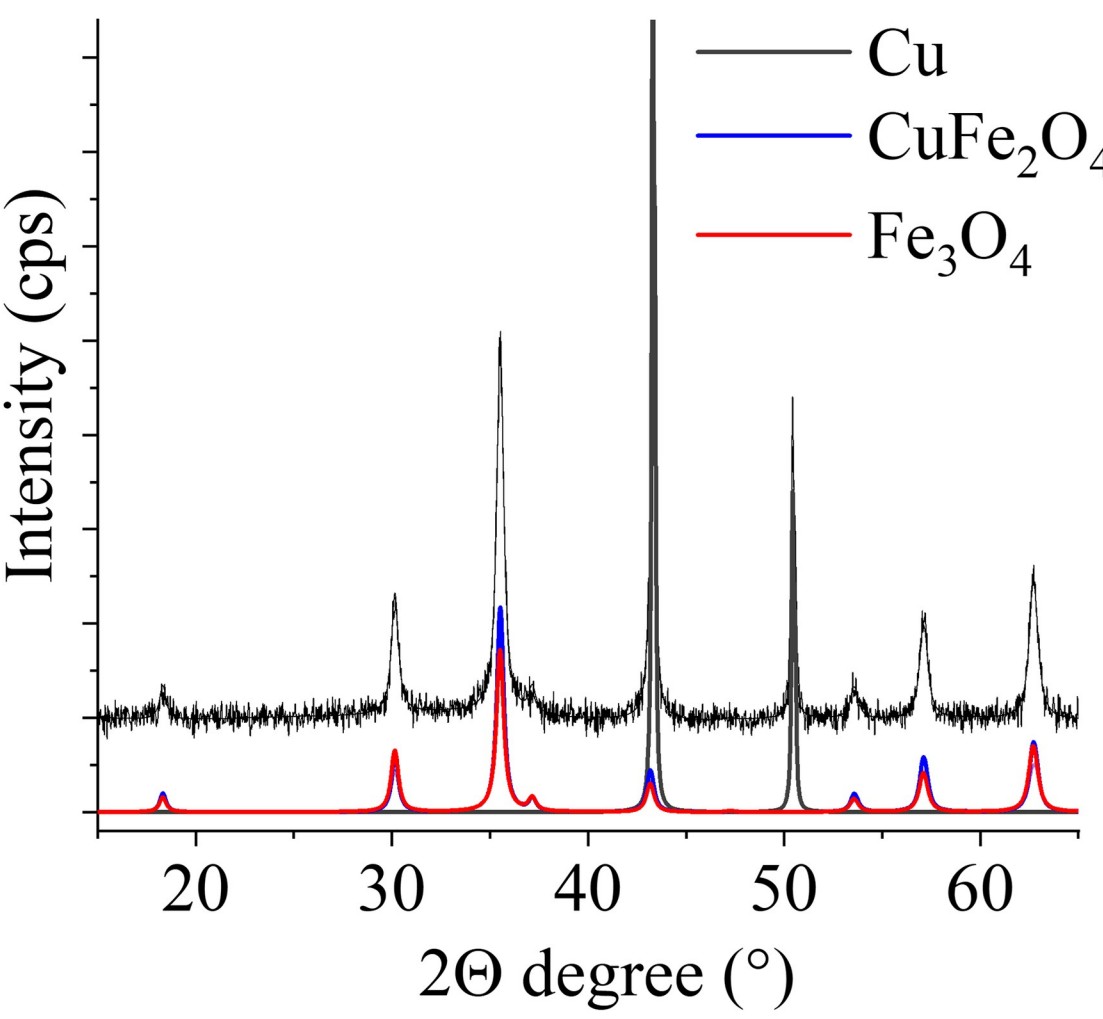

**Fig 2. Rietveld refined XRD pattern of the CuFe₂O₄-NH₂ Refl.** (A) and CuFe₂O₄-NH₂ MW (B) samples.

characteristic band of the stretching vibration mode of the metal-oxygen bond ($\nu$M-O) was found at 578 cm$^{-1}$ [69]. The peak at 879 cm$^{-1}$ belonged to the rocking vibration mode of the -CH$_2$ groups, attributable to the adsorbed ethylene glycol (EG) and monoethanolamine (MEA) molecules [69]. The stretching vibrations of $\nu$C-O and $\nu$C-N bonds resulted in bands in the wavenumber range between 1000 and 1100 cm$^{-1}$. These mentioned stretching modes belonged to the hydroxyl, carboxyl, and amine functional groups. The presence of the bending vibration of -OH groups, which caused an absorption band at 1410 cm$^{-1}$, can be explained by the adsorption of EG. The band at 1573 cm$^{-1}$ was identified as the stretching vibration of the C = C bond in the adsorbed EG and MEA molecules. The characteristic band of the amine functional groups was found at 1618 cm$^{-1}$ ($\beta$N-H) and as a shoulder on the hydroxyl band at 3225 cm$^{-1}$ ($\nu$N-H). Moreover, the band of adsorbed water molecules was visible at 1654 cm$^{-1}$ [70,71]. Vibrations indicative of organic compounds also resulted in bands at 2859 cm$^{-1}$ and 2926 cm$^{-1}$ (symmetric and asymmetric $\nu$C-H vibration modes). In the case of the CuFe$_2$O$_4$-NH$_2$ MW sample, the above-detailed vibration bands were also identified (Fig 6C). The presence of -NH$_2$ and -OH functional groups influenced the electrokinetic potentials of the nanoparticles; thus, zeta potential measurements were performed in case of the two samples. A

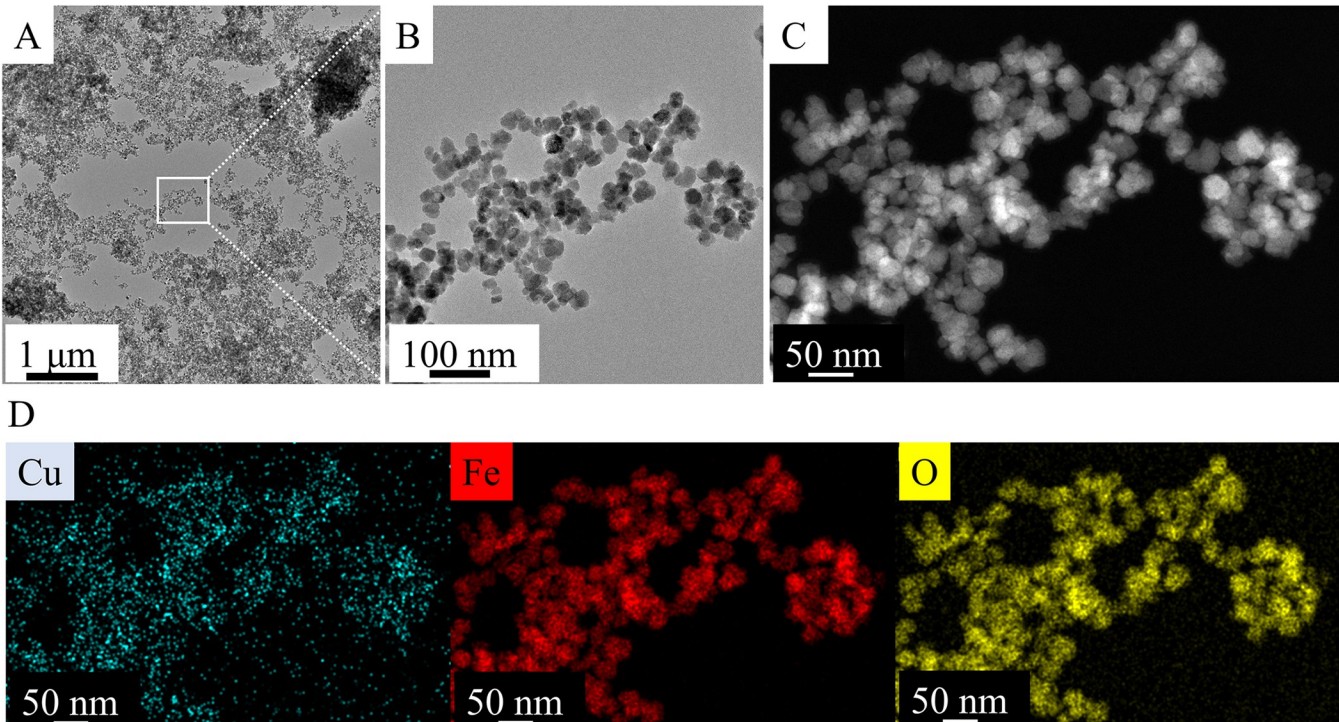

**Fig 3.** HRTEM pictures (A, B) and High-angle annular dark-field imaging (HAADF) picture (C) and element maps (D) of the CuFe$_2$O$_4$-NH$_2$ Refl. Samples.

significant difference in the zeta potentials were recorded, which could be explained by the difference in the composition and structure of nanoparticles, the averages zeta potentials were -8.6 ± 3.1 mV and + 9.2 ± 3.6 mV (Fig 6B and 6D). The measured zeta potentials were low and could not provide electrostatic colloid stability. However, in the field of imaging diagnostics, especially in the case of MRI, it is important that the colloidal stability of the contrast agents is adequate. The appropriate level of colloid stability can be achieved by steric stabilization, e.g. it can be provided by using PVP, which is a biocompatible water-soluble polymer.

### Magnetic properties of nanoparticles based on VSM measurements

For the characterization of the magnetic behavior of the two magnetic samples, VSM measurements were carried out at 298 K. The magnetization curve of the CuFe$_2$O$_4$-NH$_2$ Refl. sample showed a very narrow hysteresis loop, whereas the remanent magnetization (Mr) was 1.35 emu/g, with a coercivity (Hc) of 30 Oe. These suggest that the nano powder is ferromagnetic (Fig 7A). The magnetic saturation (Ms) was estimated from Fig 7A as 44 emu/g. In case of CuFe$_2$O$_4$-NH$_2$ MW sample the magnetization curve could not be brought to saturation even with the maximum applied magnetic field strength and hysteresis was not observed either. This refers to the superparamagnetic nature of the sample, so the saturation magnetization was calculated using the fit of the Langevin function [72]. The estimated Ms was 15 emu/g for the CuFe$_2$O$_4$-NH$_2$ MW sample, which was much lower than the value measured for copper ferrite prepared with refluxing. The copper ferrite nanoparticles from their PVP-stabilized colloids can be separated by using a magnetic field (Fig 7B and 7C).

A significant difference was observed in the saturation magnetization values of the two samples. The Ms value of the CuFe$_2$O$_4$-NH$_2$ Refl. sample was several times higher than that of the CuFe$_2$O$_4$-NH$_2$ MW sample. XRD analysis also showed that the CuFe$_2$O$_4$-NH$_2$ Refl. sample

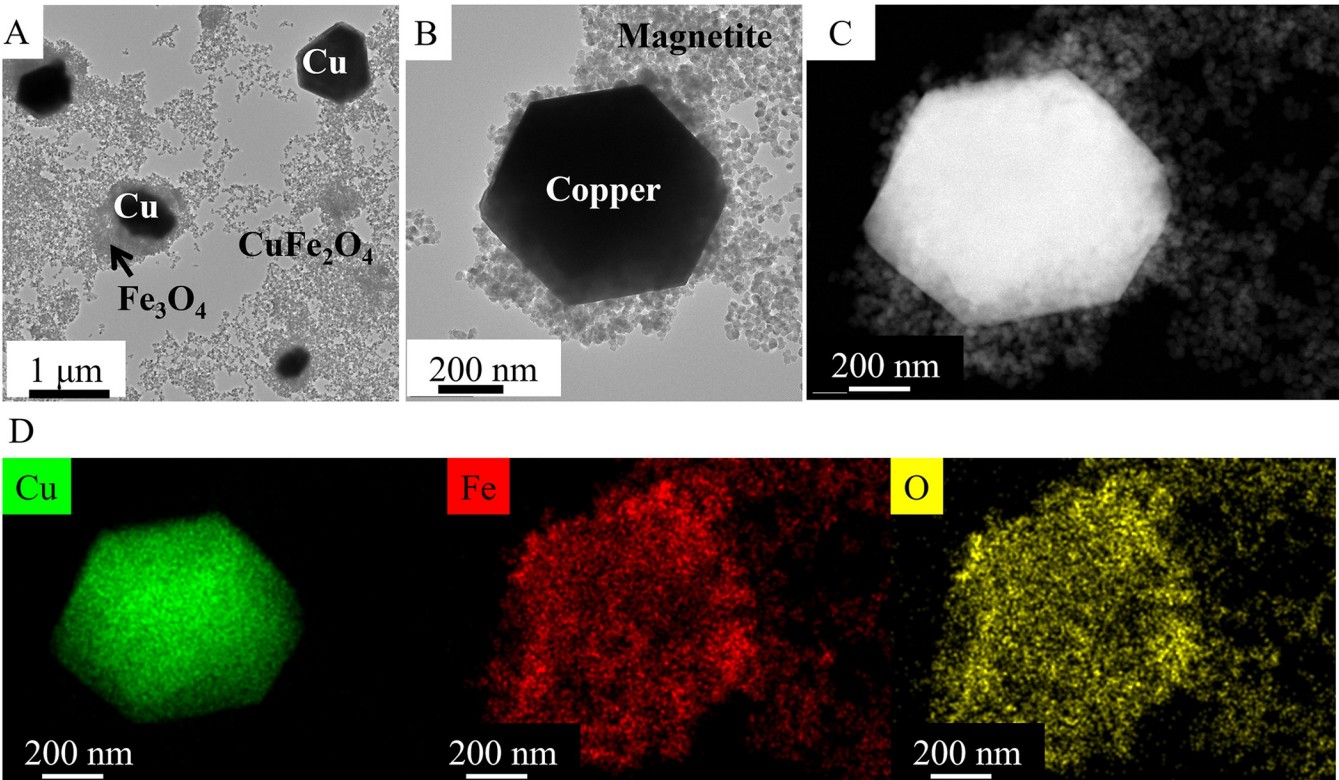

**Fig 4.** TEM (A, B) and HAADF (C) pictures of the $CuFe_2O_4$-$NH_2$ Refl. and their element maps (D).

contained a much higher amount of magnetite (32.6 wt%) than its microwave synthesized counterpart (23.2 wt%) (Table 1). Furthermore, considering that, the bulk phase magnetite has a Ms value of 90–92 emu/g, whereas the copper ferrite array phase has a lower Ms value (49–55 emu/g), it can be assumed that the quantitative relationship between the two different magnetic components also contributes to the different Ms values for the two samples produced [73,74]. The two samples show a significant difference in the size of the copper ferrite and magnetite particles (Table 1). As it is known, the saturation magnetization value decreases with decreasing particle size due to the spin disorder at the surface of the nanoparticles, which leads to the significant change of the Ms values at very small dimensions [75]. This explains the large difference in saturation magnetization values measured for two copper ferrite samples. The average sizes of the magnetic nanoparticles (magnetite and copper ferrite) were larger for the $CuFe_2O_4$-$NH_2$ Refl. sample, which leads to a higher Ms value. A similar trend was reported in the literature where 37.98 emu/g was measured with size ranging from 20 to 40 nm for $CuFe_2O_4$, 39 emu/g at an average size of 50 nm and 30 emu/g Ms in the 10–20 nm size range [76–78].

### Results of the atomic force microscopy

For the purpose of measuring and working with sub-nanometer samples, AFM is a commonly utilized imaging modality [79]. The tip convolution, which results in intentionally altered lateral dimensions on the pictures, allowed just the height of the particles to be determined during a measurement [80]. Tip convolution affected the particles' observed width. Fig 7 presents $CuFe_2O_4$-$NH_2$ Refl. and $CuFe_2O_4$-$NH_2$ MW samples as objects with a flat rectangular surface extending from a spherical halo on AFM images. While the particles' halo is the result of tip

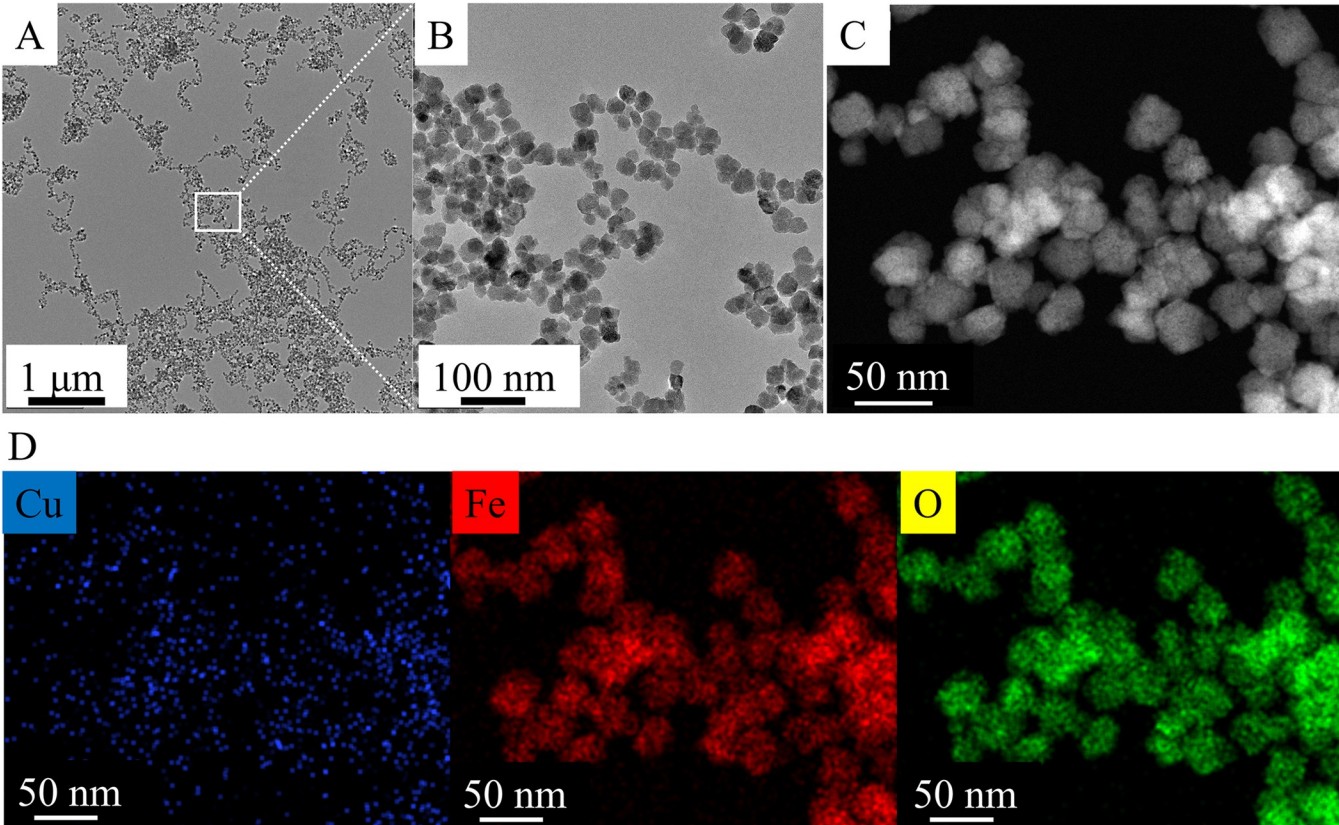

**Fig 5.** TEM (A, B) and HAADF (C) pictures of the $CuFe_2O_4$-$NH_2$ MW and their element maps (D).

convolution, that is the imaging of a rectangular prism by a tetrahedral AFM tip–the rectangular surface depicts the particles' true geometry.

In the instance of $CuFe_2O_4$-$NH_2$ Refl., the particles' rectangularity (together with their halo) was determined to be 0.83 ± 0.08 (mean ± SD), while in the case of $CuFe_2O_4$-$NH_2$ MW, it was found to be 0.72 ± 0.13 (mean ± SD). These results demonstrate that both nanoparticle species do, in fact, reflect rectangular topography (Fig 8).

### Characterization of the stability of copper ferrite colloids

The PVP-stabilized ferrites showed high stability in their aqueous colloids, it was confirmed by DLS measurements. The $CuFe_2O_4$-$NH_2$ MW samples were colloidally stable in comparison to $CuFe_2O_4$-$NH_2$ Refl. samples. DLS estimated the mean hydrodynamic diameter (intensity-based harmonic average) of $CuFe_2O_4$-$NH_2$ Refl. to be 242 ± 18 nm (average ± SD) and the CuFe2O4-NH2 MW to be 127 ± 4 nm (average ± SD). For $CuFe_2O_4$-$NH_2$ MW, this had only marginally altered over time. The results indicate that there was no substantial change in the colloidal phase over the 3.5 h of testing. The $CuFe_2O_4$-$NH_2$ MWs did not flocculate or aggregate, as indicated by the computed 0.11 ± 0.04 polydispersity index (PDI) (Fig 9).

### Results of the in vitro MRI measurements

During the in vitro MRI measurements, four different ferrite concentrations (0.02, 0.05, 0.1 and 0.2 mg/mL) in 2 mL Eppendorf tubes were used to measure the capabilities of the two samples ($CuFe_2O_4$-$NH_2$ Refl. and $CuFe_2O_4$-$NH_2$ MW) as MRI contrast agents.

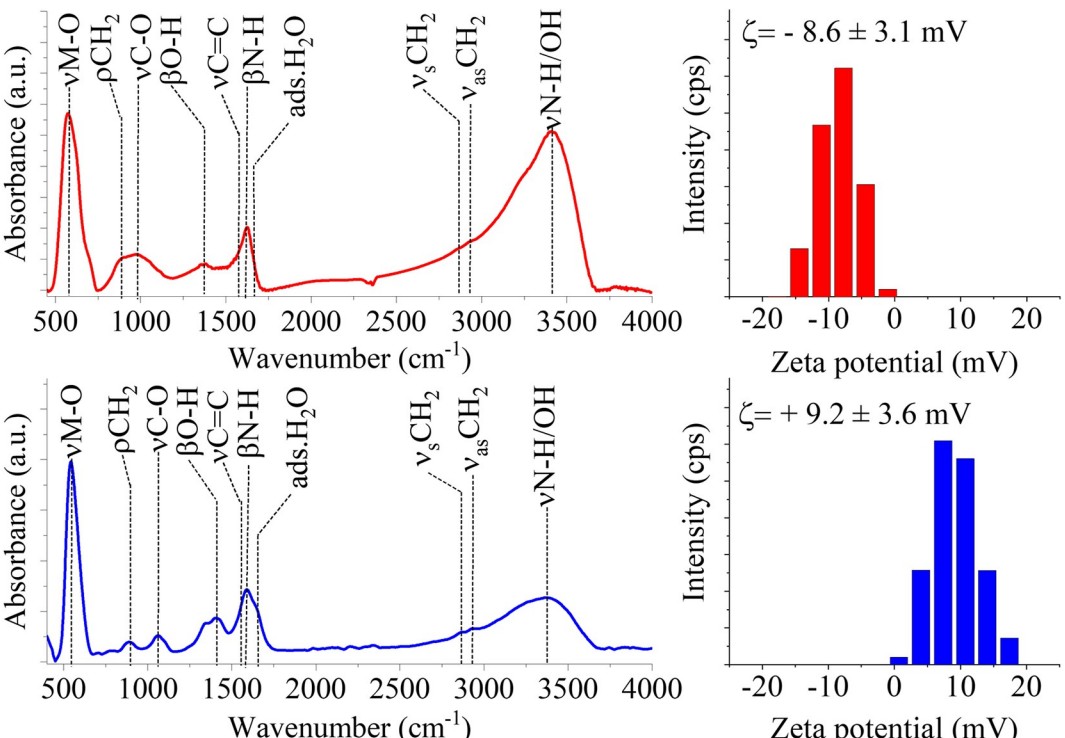

**Fig 6. FTIR spectra and zeta potential distribution of the CuFe$_2$O$_4$-NH$_2$ Refl.** (A, B) and CuFe$_2$O$_4$-NH$_2$ MW (C, D) samples.

Sedimentation would cause signal inhomogeneity on MRI scans and would prevent correct estimation of iron content in the imaging plane, making quantitative relaxivity computation impossible, which was described and detected previously in the case of ZnFe$_2$O$_4$-NH$_2$ MW particles [67]. For the CuFe$_2$O$_4$-NH$_2$ Refl. samples slight sedimentation was noted during DLS measurements, which was not observable during MRI scans. No visible difference was present between the two, examined samples (Fig 10).

Both CuFe$_2$O$_4$-NH$_2$ samples produced a homogeneous MRI signal with no evidence of sedimentation. The longitudinal relaxation times of each sample were calculated using the Multi-IR FSE scan, while the transverse relaxation times were calculated using the Multi-echo SE and Multi-echo gradient-echo (GRE) scans. All three types of relaxation times are decreasing with increasing ferrite concentration (Fig 11) as it was expected from an MRI contrast agent.

One of the most essential properties of MRI contrast agents is relaxivity, the ability to modify the relaxation time of the medium per unit concentration. The relaxivities of our samples were determined by the degree to which the relaxation rate (R1, R2, R2*), the inverse of the relaxation time, varies with ferrite concentration. Fig 11. depicts this linear relationship, and displays the fitted relaxivity values. The measured values help to observe the r2/r1 ratio and determine the "true nature" of the contrast agent: the smaller the r2/r1, the more T1 effect is present in the sample (Fig 11).

Both of the CuFe$_2$O$_4$-NH$_2$ samples (manufactured with the conventional, lengthy synthesis and the microwave-initiated synthesis) had similar characteristics to superparamagnetic iron oxide (SPIO) nanoparticles; their transverse relaxivities were in the same range as Feraheme® and Endorem®, but their r2/r1 ratio was ten times higher, suggesting that this ferrite solution was almost only a T2 contrast agent and has an undetectable T1 effect (Table 2) [81].

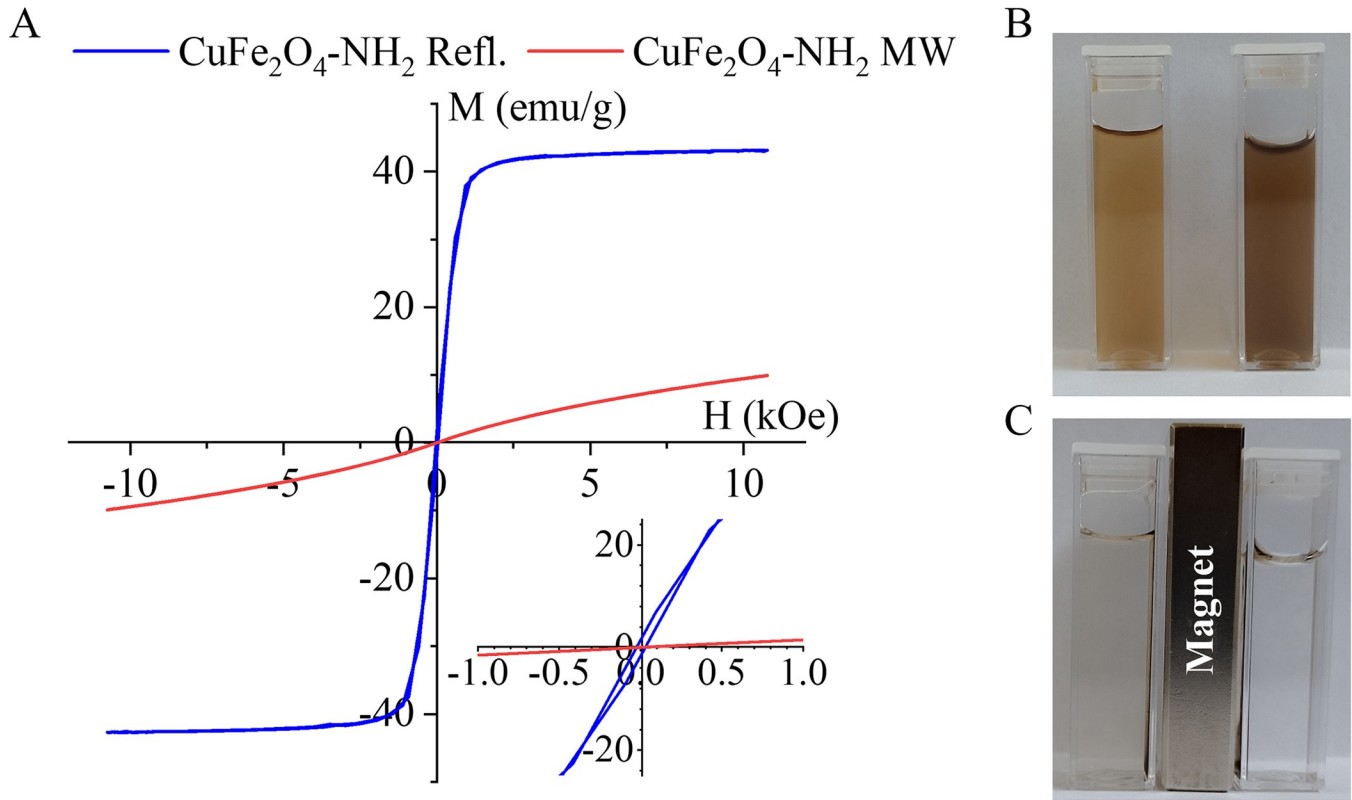

**Fig 7.** Magnetization curves (A) and magnetic separability (B and C) of CuFe$_2$O$_4$-NH$_2$ samples.

Comparing the CuFe$_2$O$_4$-NH$_2$ samples (manufactured with the conventional, lengthy synthesis and the microwave-initiated synthesis) to standard superparamagnetic iron oxide (SPIO) nanoparticles, to which they belong according to their hydrodynamic size (127 nm and 242 nm), some different characteristics had been found. The r1/r2 ratios of our samples were an order of magnitude lower, as Endorem® (with size of 80–150 nm) has r1/r2 ratio 0.044 and Resovist® (size of 62 nm) has r1/r2 ratio 0.032 [81]. It suggests that our ferite solution is almost only a T2 contrast agent and has an undetectable T1 effect (65). On the other hand, Resovist has 1.5-times as high r2 relaxivity at 3 Tesla as Endorem, the same relationship, that we found between the size and the r2 relaxivity of our samples.

### Results of the in vivo MRI measurement

Both samples were chosen to be injected in vivo, as in the in vitro measurements, both of them were stable. A concentrated sample of 1 mg/mL ferrite concentration was available for injection, from which a dose of 9.05 mg/body weight kg was intravenously injected into the tail vein as a 0.2 mL bolus. An immediate uptake was observed in the liver and in the spleen. No other organ accumulated the sample according to our T2*-weighted gradient echo scans. The Region Of Interests (ROIs) were drawn manually on the different organs and the ROI-wise change of T2*-weighted scan intensity values were determined for the first 10 min (Section A, lower on Fig 12), in which liver, kidney, vena cava and spleen uptake are clearly visualized.

According to the DLS measurement, the hydrodynamic size of CuFe$_2$O$_4$-NH$_2$ particles is in the range of SPIO nanoparticles, and the biodistribution of both are also similar. These

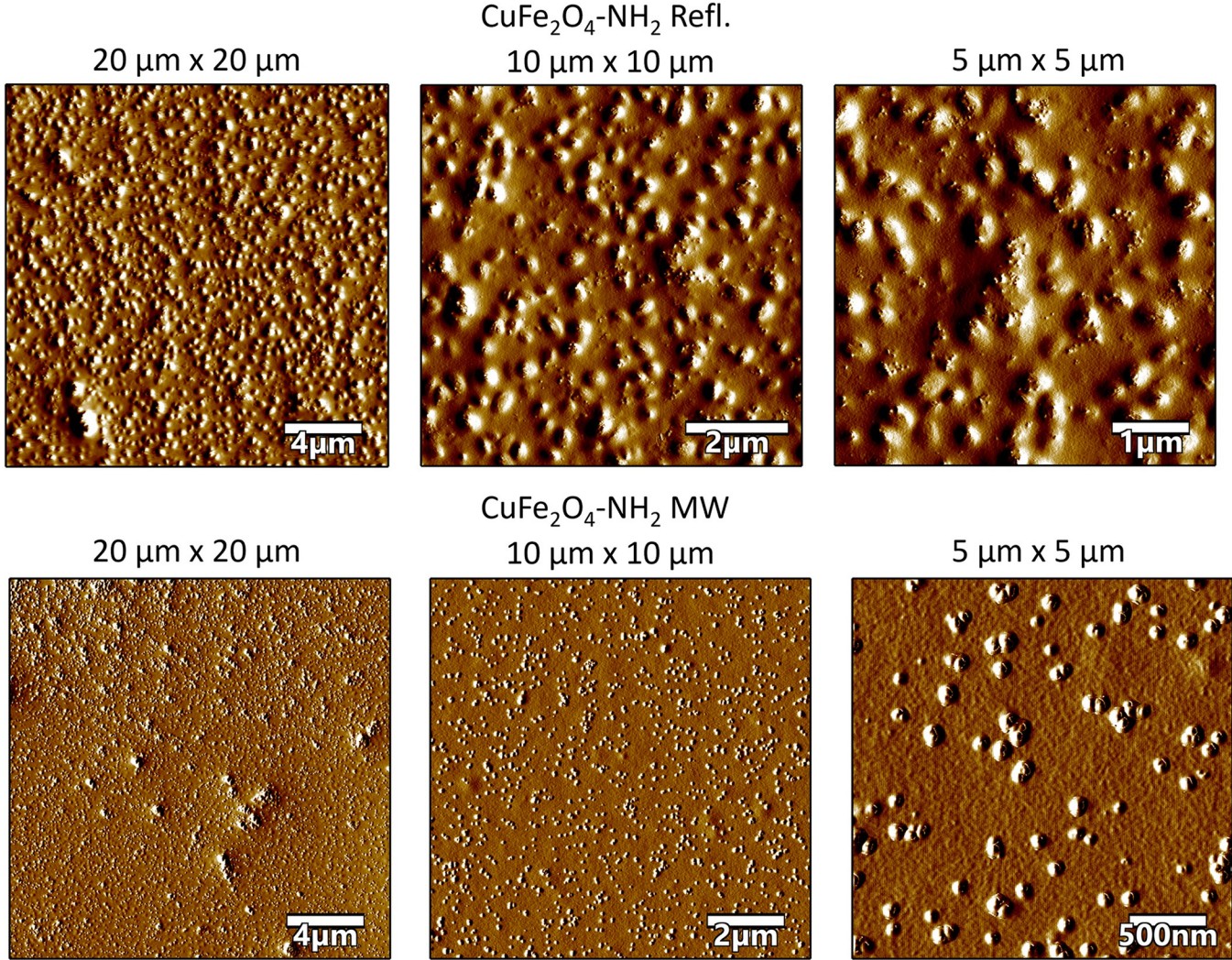

**Fig 8. Atomic force microscopy (AFM) amplitude-contrast images with different magnification of CuFe$_2$O$_4$-NH$_2$ samples on mica surface.** The nanoparticles are presented as spherical objects, covered with the PVP polymer on their surface. The size of the images is 20 μm x 20 μm, 10 μm x 10 μm (in case of CuFe$_2$O$_4$-NH$_2$ Refl. sample) and 5 μm x 5 μm respectively.

parameters suggest also similar pharmacodynamic effects in vivo. For application in high-quality contrast MRI, the relatively long blood circulation time is essential, provided by the appropriate particle size. In this case, the particles are similarly prepared as the previously presented ZnFe$_2$O$_4$-NH$_2$ samples, long retention times can be expected in the liver too [67,82].

## Conclusion

NH$_2$-functionalized copper ferrite nanoparticles were synthesized by a co-precipitation method in organic solvent, using two different synthesis techniques. The conventional method, which involves a 12 h reflux step, is a time and energy consuming synthesis route. This process was modified by MW assisted synthesis instead of a 12 h reflux. The microwave synthesis was much faster and more economical, requiring only 4 minutes to form the ferrite nanoparticles. The morphological characteristics of the two ferrite samples were different, as

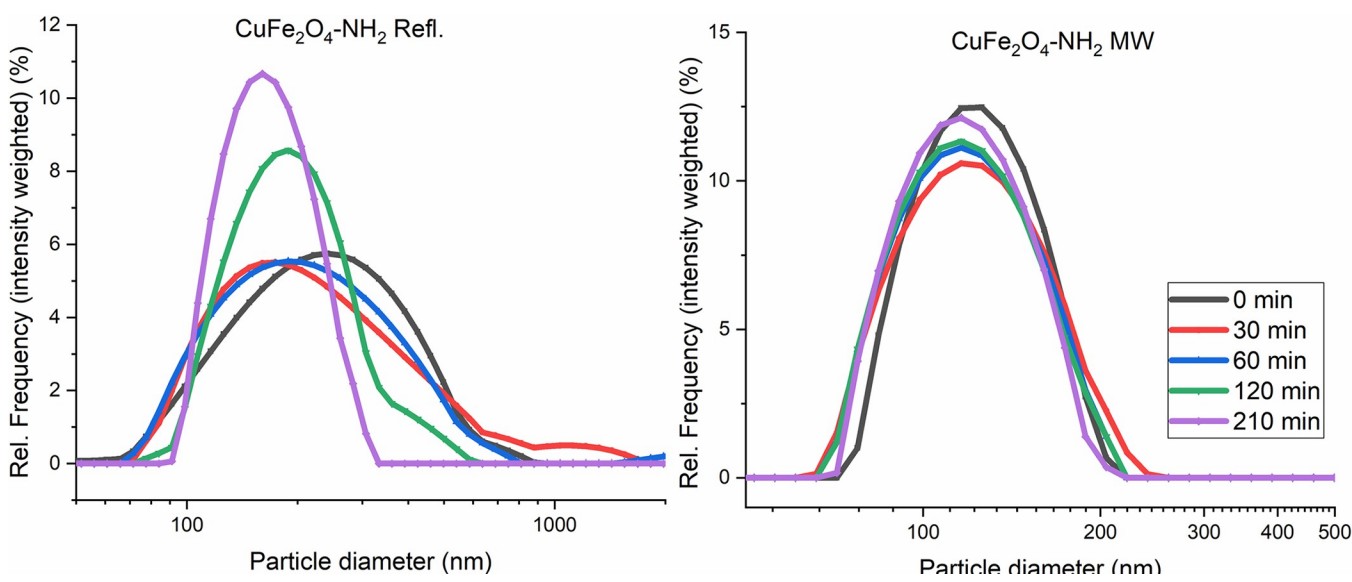

**Fig 9. The size-distribution shift over time in the CuFe$_2$O$_4$-NH$_2$ Refl. and CuFe$_2$O$_4$-NH$_2$ MW samples.** Based on these results, CuFe$_2$O$_4$-NH$_2$ MW has "better" colloidal stability over time.

large copper nano- and microplates were formed during the reflux step (CuFe$_2$O$_4$-NH$_2$ Refl.). In the CuFe$_2$O$_4$-NH$_2$ MW samples the abovementioned copper particles were not detected. In the case of the two samples, copper ferrite and magnetite were also found, based on phase identification measurements (XRD). In addition, the magnetic properties, based on VSM measurements, showed a significant difference. Which can be identified in the zeta potential values of the two samples. The CuFe$_2$O$_4$-NH$_2$ Refl. samples had a zeta potential of -8.6 ± 3.1 mV, on the contrary, the CuFe$_2$O$_4$-NH$_2$ MW sample +9.2 ± 3.6 mV was measured. The longer solvothermal synthesis resulted in the formation of more negatively charged ferrite particles. Due to the low zeta potentials of the nanoparticles, the stability of their aqueous colloidal systems was not stable. To overcome the problem of colloidal stability, these ferrite nanoparticles were stabilized by polyvinyl pyrrolidone in a dried form. The copper ferrite nanoparticles were embedded in PVP (ferrite content of 50 mg/g). These polymer-stabilized red-brown crystals could be easily redispersed in distilled water. Two PVP-coated CuFe$_2$O$_4$ samples were compared as contrast agents in magnetic resonance imaging (MRI).

To conclude, the comparison analysis of CuFe$_2$O$_4$-NH$_2$ Refl. and CuFe$_2$O$_4$-NH$_2$ MW nanoparticles clearly indicated that the novel method is more beneficial. CuFe$_2$O$_4$-NH$_2$ MW is excellent performance as an MRI contrast agent is further demonstrated by its exceptional colloidal stability and uniformly homogenous MRI signals at multiple ferrite concentrations. This advantage is particularly noticeable when compared to CuFe$_2$O$_4$-NH$_2$ Refl., where potential problems with sedimentation during DLS measurements were faced, where potential problems with sedimentation and reduced signal homogeneity during in vitro MRI tests were faced.

These results highlight the critical role of the synthesis techniques play in determining the characteristics of nanoparticles, emphasizing the benefits of ecologically friendly strategies such as the thermal treatment method. This work shows the practical advantages of choosing the best species of nanoparticle for a given diagnostic application, while also offering insightful information about the subtle characteristics of copper ferrite nanoparticles. The exceptional colloidal stability and imaging properties of CuFe$_2$O$_4$-NH$_2$ MW present intriguing

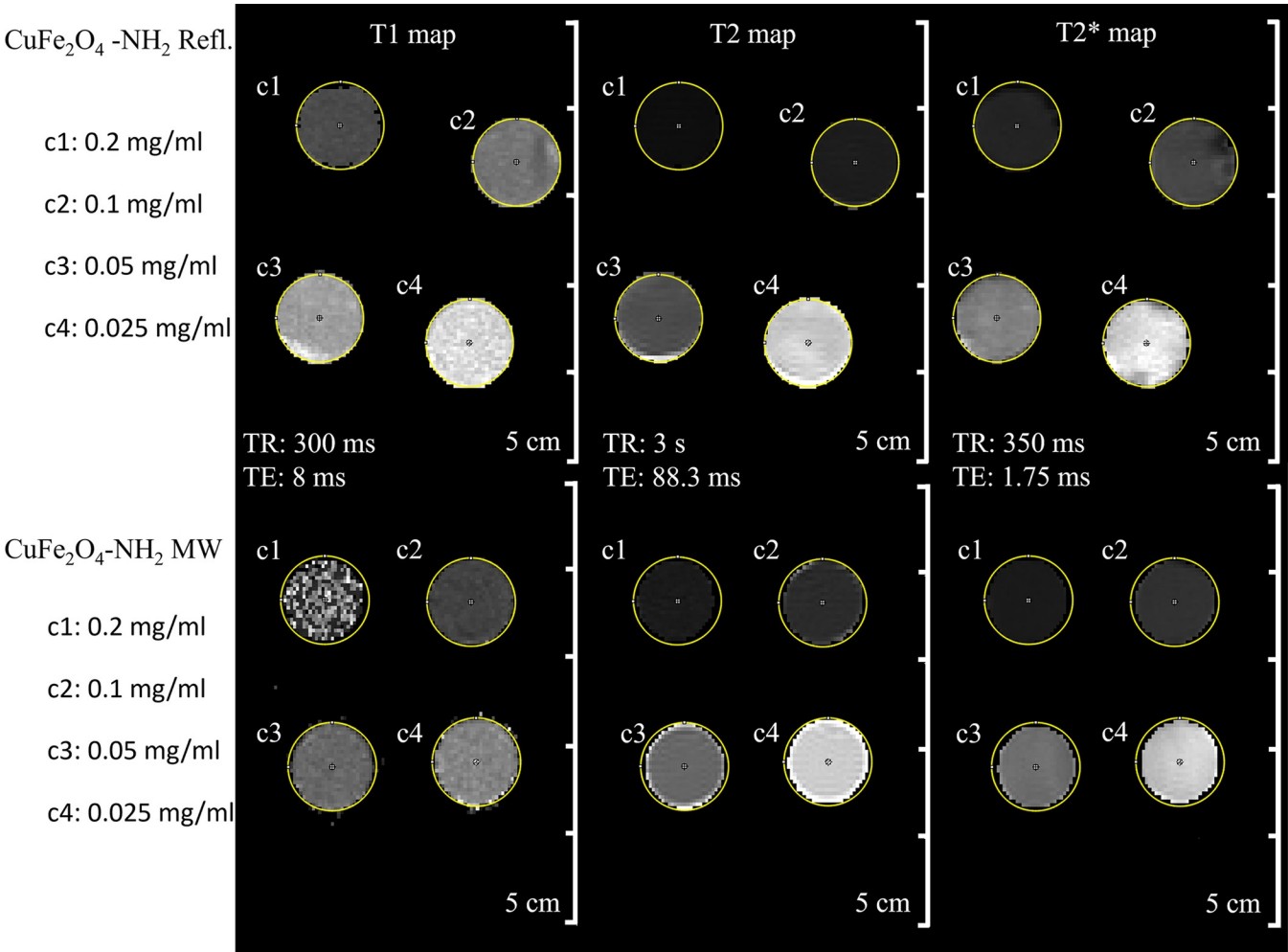

**Fig 10. Summary of the produced quantitative MRI maps in the aqueous dispersion of the CuFe₂O₄-NH₂ samples in different dilutions.** The T1-maps were based on Multi-IR fast spin echo (FSE) scan, the T2-maps were based on Multi-echo FSE scan and the T2*-maps were based on Multi-echo GRE scan of the ferrite sample in different dilutions.

opportunities for MRI improvements that could have a substantial influence on the area of diagnostic imaging. Furthermore, based on the results presented, the refluxing process can be used to successfully produce particles with a core-shell structure, featuring copper particles inside and nanoparticles on the outside. The excellent thermal conductivity of the copper raises the possibility that these core-shell particles could be used in magnetic hyperthermia treatment for cancer.

**Table 2. Longitudinal (r1) and transversal (r2 and r2*) relaxivity of CuFe₂O₄-NH₂ samples.**

|                    | CuFe₂O₄-NH₂ Refl. | CuFe₂O₄-NH₂ MW |
|--------------------|-------------------|----------------|
| r2* (mL/mg/ms)     | 1.028             | 1.496          |
| r2 (mL/mg/ms)      | 0.691             | 1.127          |
| r1 (mL/mg/ms)      | 0.002             | 0.006          |

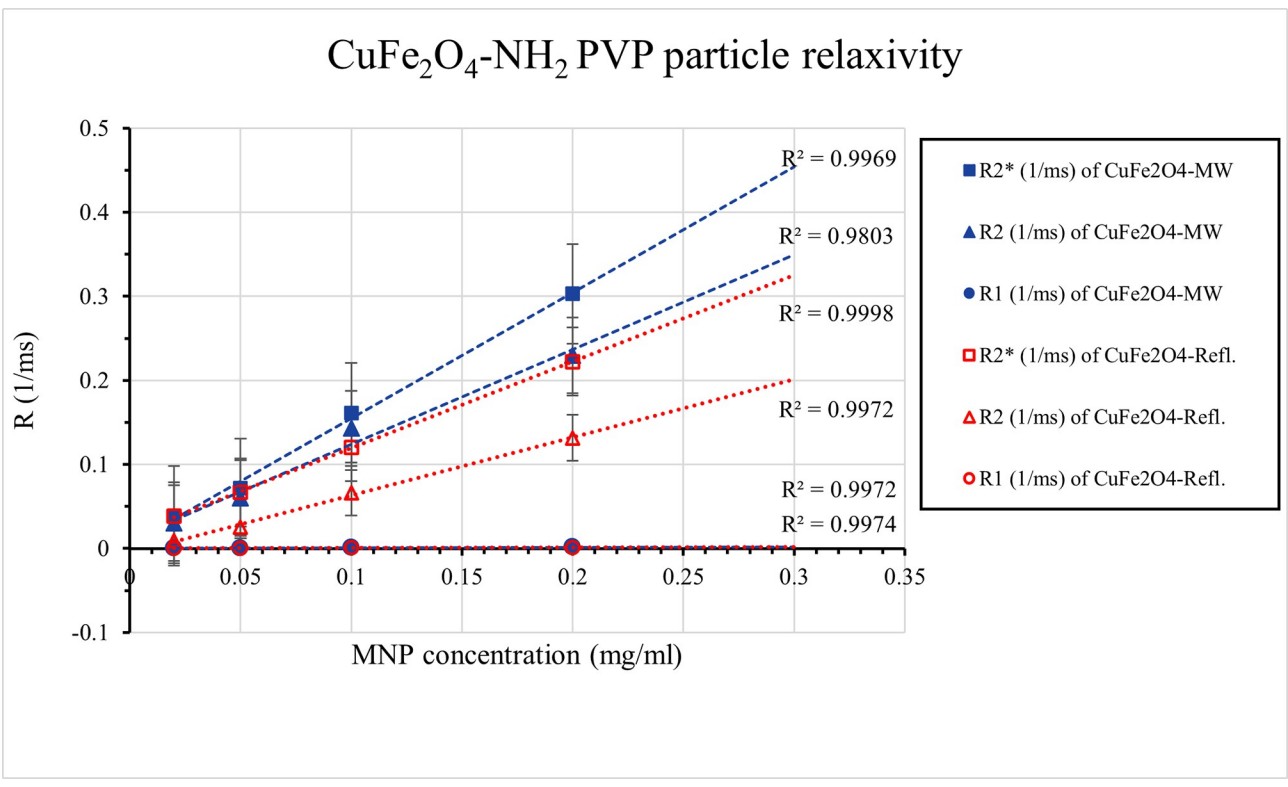

**Fig 11. Plot of transversal and longitudinal relaxation time changes with iron concentration.** Linear fit was used for relaxivity calculations.

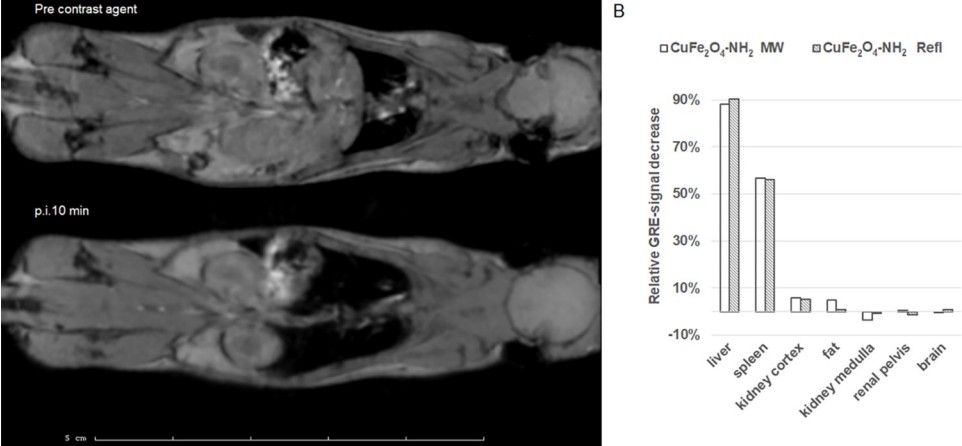

**Fig 12. T2\*-weighted gradient echo (GRE) scans of a mouse at 2 different timepoints—before injection and 10 min after intravenous injection of CuFe$_2$O$_4$-NH$_2$ samples (A image).** The liver, spleen and kidney cortex are highlighted (A image). The ROI-wise relative change of T2\*-weighted intensities (B image) shows the highest uptake in the liver. T2\* contrast refers to the decay of transverse magnetization seen with gradient-echo (GRE) sequences. Identical signal-intensity levels were measured in the case of both samples.

## Supporting information

**S1 Data. Raw measurement data for in vitro and in vivo MRI, DLS and preliminary results for Alamar Blue toxicology assay are available in the attached supplemental info raw data.** (ODS)

## Author Contributions

**Conceptualization:** Fatemeh Heydari, Ágnes M. Ilosvai, Ferenc Kristály, Lajos Daróczi, Béla Viskolcz, László Vanyorek, Krisztián Szigeti.

**Data curation:** Fatemeh Heydari, Noémi Kovács, Ferenc Kristály, Lajos Daróczi, Zoltán Kaleta, Miklós Nagy, László Vanyorek.

**Formal analysis:** Fatemeh Heydari, Noémi Kovács, Ferenc Kristály, Lajos Daróczi, Zoltán Kaleta, Miklós Nagy, László Vanyorek.

**Funding acquisition:** Domokos Máthé, Béla Viskolcz, László Vanyorek, Krisztián Szigeti.

**Investigation:** Fatemeh Heydari, Ágnes M. Ilosvai, Noémi Kovács, Lajos Daróczi, Zoltán Kaleta, László Forgách.

**Methodology:** Ágnes M. Ilosvai, Ferenc Kristály, Béla Viskolcz, Miklós Nagy, László Vanyorek, László Forgách, Krisztián Szigeti.

**Project administration:** Fatemeh Heydari, Béla Viskolcz, Miklós Nagy, László Vanyorek, László Forgách, Krisztián Szigeti.

**Resources:** Domokos Máthé, Zoltán Kaleta, László Vanyorek, László Forgách, Krisztián Szigeti.

**Software:** Ágnes M. Ilosvai, Noémi Kovács, Ferenc Kristály.

**Supervision:** Domokos Máthé, Krisztián Szigeti.

**Validation:** Noémi Kovács, Domokos Máthé, Ferenc Kristály, Lajos Daróczi, Béla Viskolcz, Miklós Nagy, László Forgách, Krisztián Szigeti.

**Visualization:** Ágnes M. Ilosvai.

**Writing – original draft:** Fatemeh Heydari, Ágnes M. Ilosvai, Domokos Máthé, Zoltán Kaleta, Béla Viskolcz, Miklós Nagy, László Vanyorek, László Forgách, Krisztián Szigeti.

**Writing – review & editing:** Ágnes M. Ilosvai, Domokos Máthé, Zoltán Kaleta, Béla Viskolcz, Miklós Nagy, László Vanyorek, László Forgách, Krisztián Szigeti.

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
