## [Decision Letter · Decision Letter 0]

5 Sep 2024

PONE-D-24-34021Solvothermal synthesis of polyvinyl pyrrolidone encapsulated, amine-functionalized copper ferrite and its use as a Magnetic Resonance Imaging contrast agentPLOS ONE

Dear Dr. Forgách,

Thank you for submitting your manuscript to PLOS ONE. After careful consideration, we feel that it has merit but does not fully meet PLOS ONE’s publication criteria as it currently stands. Therefore, we invite you to submit a revised version of the manuscript that addresses the points raised during the review process.

We look forward to receiving your revised manuscript.

Kind regards,

Amir Elzwawy, Ph.D.

Academic Editor

PLOS ONE

Journal requirements: 1. When submitting your revision, we need you to address these additional requirements. Please ensure that your manuscript meets PLOS ONE's style requirements, including those for file naming. The PLOS ONE style templates can be found at https://journals.plos.org/plosone/s/file?id=wjVg/PLOSOne_formatting_sample_main_body.pdf and https://journals.plos.org/plosone/s/file?id=ba62/PLOSOne_formatting_sample_title_authors_affiliations.pdf. 2. We noticed you have some minor occurrence of overlapping text with the following previous publication(s), which needs to be addressed: - Development of Polymer-Encapsulated, Amine-Functionalized Zinc Ferrite Nanoparticles as MRI Contrast Agents (doi:  10.3390/ijms242216203)(among others)  In your revision ensure you cite all your sources (including your own works), and quote or rephrase any duplicated text outside the methods section. Further consideration is dependent on these concerns being addressed. 3. To comply with PLOS ONE submissions requirements, in your Methods section, please provide additional information regarding the experiments involving animals and ensure you have included details on (1) methods of sacrifice, (2) methods of anesthesia and/or analgesia, and (3) efforts to alleviate suffering. 4. Funding Information and Financial Disclosure sections do not match:We note that the grant information you provided in the ‘Funding Information’ and ‘Financial Disclosure’ sections do not match.  When you resubmit, please ensure that you provide the correct grant numbers for the awards you received for your study in the ‘Funding Information’ section. 5. Thank you for stating the following financial disclosure:  [This research was supported by the National Research, Development, and Innovation Fund (Hungary), within the TKP2021-NVA-14 project. F.H. received funding withing the framework of the TKP2021-EGA23. This work was supported by The European Union’s Horizon 2020 Research And Innovation Program, grant agreements No. 739593: HCEMM, supported by the EU Programme: H2020-EU.4.a. and No. 859890 SmartAge ITN. Part of the work was also supported in the cadre of the iMatrix consortium at H2020-JTC2022 ERA-NET-Neuron. This work was also partly funded by grants from the Hungarian National Research, Development, and Innovation Office (Thematic Excellence Program, TKP-BIOImaging, financed under the 2020-4.1.1-TKP2020 funding scheme, Investment to the Future 2020.1.16-Jöv˝o-2021-00013 and TKP2021-EGA-23).].  Please state what role the funders took in the study.  If the funders had no role, please state: ""The funders had no role in study design, data collection and analysis, decision to publish, or preparation of the manuscript."" If this statement is not correct you must amend it as needed. Please include this amended Role of Funder statement in your cover letter; we will change the online submission form on your behalf. 6. We note that you have indicated that there are restrictions to data sharing for this study. PLOS only allows data to be available upon request if there are legal or ethical restrictions on sharing data publicly. For more information on unacceptable data access restrictions, please see http://journals.plos.org/plosone/s/data-availability#loc-unacceptable-data-access-restrictions.  Before we proceed with your manuscript, please address the following prompts: a) If there are ethical or legal restrictions on sharing a de-identified data set, please explain them in detail (e.g., data contain potentially identifying or sensitive patient information, data are owned by a third-party organization, etc.) and who has imposed them (e.g., a Research Ethics Committee or Institutional Review Board, etc.). Please also provide contact information for a data access committee, ethics committee, or other institutional body to which data requests may be sent. b) If there are no restrictions, please upload the minimal anonymized data set necessary to replicate your study findings to a stable, public repository and provide us with the relevant URLs, DOIs, or accession numbers. For a list of recommended repositories, please seehttps://journals.plos.org/plosone/s/recommended-repositories. You also have the option of uploading the data as Supporting Information files, but we would recommend depositing data directly to a data repository if possible. We will update your Data Availability statement on your behalf to reflect the information you provide. 7. In the online submission form, you indicated that [All relevant data are within the manuscript. The raw measurement data is only available upon request from the authorts]. All PLOS journals now require all data underlying the findings described in their manuscript to be freely available to other researchers, either 1. In a public repository, 2. Within the manuscript itself, or 3. Uploaded as supplementary information.This policy applies to all data except where public deposition would breach compliance with the protocol approved by your research ethics board. If your data cannot be made publicly available for ethical or legal reasons (e.g., public availability would compromise patient privacy), please explain your reasons on resubmission and your exemption request will be escalated for approval. 

Additional Editor Comments:

The authors are requested to provide a detailed response to the reviewers. Multiple concerns are requested and I'm providing a major revision opportunity after considering all opinions of the reviewers.

Pay attention to all comments especially reviewer3. If the response is not fully addressed to all reviewers, the rejection possibility might occur.

with regards

Reviewers' comments:

Reviewer's Responses to Questions

**Comments to the Author**

1. Is the manuscript technically sound, and do the data support the conclusions?

Reviewer #1: Yes

Reviewer #2: Yes

Reviewer #3: Partly

Reviewer #4: Yes

Reviewer #5: Yes

Reviewer #6: Yes

2. Has the statistical analysis been performed appropriately and rigorously? 

Reviewer #1: Yes

Reviewer #2: Yes

Reviewer #3: No

Reviewer #4: N/A

Reviewer #5: Yes

Reviewer #6: Yes

3. Have the authors made all data underlying the findings in their manuscript fully available?

Reviewer #1: Yes

Reviewer #2: Yes

Reviewer #3: No

Reviewer #4: Yes

Reviewer #5: Yes

Reviewer #6: Yes

4. Is the manuscript presented in an intelligible fashion and written in standard English?

Reviewer #1: No

Reviewer #2: Yes

Reviewer #3: No

Reviewer #4: No

Reviewer #5: No

Reviewer #6: No

5. Review Comments to the Author

Reviewer #1: 1. While the abstract mentions valuable insights and applications, it could further emphasize the broader implications of this research. How does this advancement contribute to the field of MRI or materials science in general?

o Grammar and Word Choice:

o Line 29: There are some grammatical errors, such as "improved porperties in case of the of CuFe2O4-NH2 MW particles" which should be corrected to "improved properties in the case of the CuFe2O4-NH2 MW particles." The use of “of” twice in that sentence appears to be a typo.

o Some sentences could be made more concise. For instance, the phrase "an innovative method was successfully applied and has yielded a novel particle with enhanced properties compared to traditional ferrite-based materials" could be streamlined. Consider rephrasing it to "This innovative method successfully produced nanoparticles with enhanced properties compared to traditional ferrite materials."

2. Materials:

o Please write the chemical formula correctly using subscripts.

o Ensure you consistently use the same term (e.g., "ethanolamine" vs "ethanol amine") to avoid confusion.

o Ensure consistency in the naming conventions used for compounds and solvents throughout the document. Example: Polyvinylpyrrolidone (PVP)

o Line 116: “20 ml of ethanol amine was added” Please convert this to mmol.

o Line 116: The phrase "continuous continuous mixing" seems to be a typographical error; use "continuous stirring”

3. Review of Results and Discussion

o The section describes the characterization of amine-functionalized magnetic nanoparticles synthesized through two different methods: reflux and microwave-assisted methods (CuFe2O4-NH2 Refl. and CuFe2O4-NH2 MW). Key aspects of the results are highlighted, including the crystallographic phases identified via X-ray diffraction (XRD) and specific phase compositions derived from Rietveld refinement.

o The paragraph could benefit from clearer structuring. Consider breaking up long sentences for better readability. For instance, the sentence about the presence of copper peaks could be split to enhance clarity.

o Use subheadings where appropriate (e.g., “XRD Analysis” or “Particle Size Distribution”) to help readers navigate the section easily.

o Line 206: "mesarured" which should be "measured," "

o Provide a brief explanation or background on why the determination of these phases (CuFe2O4, magnetite, Cu) is significant for the application as MRI contrast agents. This context can help readers understand the relevance of the results.

4. Ensure all statements regarding background information and previous research are properly cited.

5. Minor grammatical errors and sentence structure issues could be addressed for a more polished presentation. The overall tone is professional, but a thorough proofreading would enhance clarity.

6. Incorporating an MTT assay in your study of copper ferrite nanoparticles would be a strong choice to assess the cytotoxicity, provided it is part of a broader strategy that may include additional assays for validation.

7. A comparison with other published work must be performed to contextualize the findings and assess their significance concerning existing literature on similar nanoparticles." Especially with your published article https://doi.org/10.3390/ijms242216203 what is the impact of Cu compared to Zn

8. The conclusions section should give the importance and usefulness of the research results

Reviewer #2: Add miller indices in xrd patterns

Add error bar on statistical curve

Compeare each obtaoned data with similar work

The reason of using each precursor should be mentioned in exp. Section

Conclusion should be wrtten more comprehensive

Reviewer #3: The paper „Solvothermal synthesis of polyvinyl pyrrolidone encapsulated, amine-functionalized copper ferrite and its use as a Magnetic Resonance Imaging contrast agent“, written by Fatemeh Heydari et al., presents copper ferrite nanoparticles for contrast-enhanced MRI and a novel method of their preparation.

The authors present an improved synthesis method and compare nanoparticles prepared by a standard solvothermal synthesis and modified, microwave-assisted procedure. Particles prepared by both ways were compared by a row of physical measurements (magnetometry, X-ray diffraction, dynamic light scattering, relaxometry), and tested contrast enhanced MRI in vitro and in vivo.

However, it is not clear, what is the main message: new particles for MR imaging, or the new procedure for preparation? Incorporation of copper into the nanoparticles might be interesting, but not from the point of view of MR imaging. Standard commercially available nanoparticles based on iron oxides have better performance (same or higher r2 relaxivity, substantially higher r1 relaxivity). If there is no special function of the copper in the nanoparticles (or of the particles themselves), there is no reason to use them instead of standard iron oxide nanoparticles. Therefore, I suppose, that the authors should emphasize the positive point of their work – the novel preparation method, thoroughly discuss the physical properties of nanoparticles prepared in a new way, discuss possible use of the method for preparation other nanoparticle types, and possibly think of submitting the paper to more technically oriented journal.

I suppose that it should be rather rewritten than revised. I am sorry.

Specifically:

I appreciate a row of physical methods for characterization of nanoparticles; nevertheless, the paper suffers of many weak parts:

Language: I am not a native speaker, but I dare to say that the manuscript requires a thorough proofread. Many typos, mistakes or strange sentences strike even me. There are also typographic errors (physical quantities should be in italics, the authors should use subscript for numbers in chemical formulas, superscript for exponents etc.) throughout the text.

Also, I think that there are wrong references to the images in the text.

Point by point comments:

p. 3, l. 29: There should be plural of the verb “show”, I suppose. Typo in “properties”

p. 3, l. 30-32: Word order is strange.

p. 4, l. 77-81: I have to remind, that NFS was extremely rare and since identifying the cause of NSF and adoption of relevant measures (such as avoiding application of linear Gd chelates to patients with renal failure), no case of NFS related to Gd contrast agents was reported. On the other hand, iron-oxide nanoparticles have much more frequent side effects (albeit not fatal) and there is no nanoparticle-based contrast agent currently approved for clinical use in Europe. Therefore, dreams of replacing Gd-based contrast agents by magnetic nanoparticles in near future are just wishful thinking (I do not oppose to study and explore magnetic nanoparticles, they are for sure very interesting and useful, but, please, do not defame Gd contrast agents, which are currently irreplaceable and I believe that they will be replaced by novel native MRI sequences rather than by nanoparticles).

p. 4, l. 95-96: “Capping” agents or molecules is quite strange, I would recommend “coating”

p. 5, l. 117: Please, use RCF instead of RPM

p. 6, l. 133: “Techniques”, not “technics”

p. 8, l. 189-190: In vivo experiments are insufficiently described. How many mice the authors used? Which strain? Age? Sex? How was the suspension applied? Volume? Concentration? (Part of this information is in the Result section, but it should be here!)

p. 8, l. 190-193: The described sequence (with a 90° excitation pulse and TR = 270 ms) has also quite strong T1-weighting, I suppose. Please, comment whether it may affect the results.

Which organs were examined? How was the signal calculated/compared?

p. 9: Figure 1 has no caption. Nevertheless, I think this figure is redundant.

p. 9, l. 214: The caption to Fig. 2 refers to A and B figures, however, there is only one graph in Fig. 2.

p. 10, Table 1: How many crystallites was evaluated to obtain average size (should be described in Methods)

p. 10-11: What is the difference between Fig. 3 and 4? The caption is almost the same and it shows the same type of nanoparticles. Please explain.

p.12, l. 268: “spectra”, not “spectrums”

p. 12, l. 272-273: The authors describe non-zero remanent magnetization of the nanoparticles and conclude that “the nano powder is paramagnetic in nature”. Existing remanent magnetization contradicts paramagnetism!

p. 12, l. 270-279: There is a substantial difference in saturation magnetization between the two nanoparticle types (Ms of CuFe2O4-NH2 Refl is three times higher than Ms of CuFe2O4-NH2 MW), although they are chemically identical. I miss there an explanation or at least a hypothesis. This is crucial for the nanoparticle performance.

p. 12, l. 278: The statement “The PVP-stabilized ferrites showed high stability in their aqueous colloids“ might be correct, however, experiments proving it are described later in the paper, therefore, save it for following paragraphs. Does the colloid redisperse itself when it is magnetized (see Fig. 7C)?

p. 12, l. 282-283: The sentence is incomprehensible to me. How can you get concentration 1 mg/mL by dispersing of 100 mg in 5 mL???

p. 13, l. 285-294: I am puzzled by the terms of “rectangularity”, however, I am not familiar with AFM, so it may be just lack of knowledge on my side.

p. 13, l. 301-306: The colloid stability was really measured within three hours only? It is insufficient for in vivo applications!

p. 14, l. 311-321: This part are not results. Explanation of relaxation times in MRI and their role in clinical practice belongs rather to the Introduction. Following sentence belongs to the Methods.

p. 14, l. 328: Reference to Fig. 9 should be probably Fig. 10.

p. 15, l. 336: Fig. 11 instead of 10?

p. 15, l. 339: Fig. 11 instead of 10?

p. 15, l. 345-350: The authors conclude that “relaxivities were in the same range as Feraheme and Endorem”, I would expect somewhat more precise comparison. It requires some calculation, as relaxivities are usually presented in [s-1/mM]. Recalculated r2 for CuFe2O4-NH2 Refl is 58.8 s-1/mM, and for CuFe2O4-NH2 MW 95.9 s-1/mM (the coating was neglected – by the way, it is not clear, whether the measurements were performed on coated particles), while Endorem has r2 = 158 s-1/mM and Ferahem 83 s-1/mM (both at lower field 1.5 T, so the values will be even higher at 3 T), see e.g., http://dx.doi.org/10.1055/s-0033-1335438

Also, I would expect substantially higher relaxivity in the case of CuFe2O4-NH2 Refl, which has three times higher saturation magnetization and much steeper magnetization curve than CuFe2O4-NH2 MW. Can you comment on this?

p. 15, l. 351: I miss an information about the safety of the nanoparticles prior to in vivo application to experimental animals. Did authors perform cytotoxicity tests? Did they test the long-term colloidal stability? Is the PVP coating stable? PVP itself is soluble in water!

p. 16, l. 354-358: The authors state that “An immediate uptake was observed in the liver and in the spleen. No other organ accumulated the sample according to our T2*-weighted gradient echo scans.“ However, on line 357, they write that “…liver, kidney, vena cava and spleen uptake are clearly visualized“. Please, correct.

Formal omissions:

Funding should include initials of the authors who received each award (required by the publisher)

Missing explanation, why the data are not available („Stating ‘data available on request from the author’ is not sufficient. If your data are only available upon request, select ‘No’ for the first question and explain your exceptional situation in the text box.“)

Reviewer #4: 1. Characterization technics is wrongly written. please check the spellings. In introduction section, I suggest authors to add some more information about substitution . In the introduction, there is a lack of application; at the same time, some recent technological developments are missing. Strongly mention the novelty of the paper and why do authors choose this method? Also, please highlight what's new in yours

2. The authors must highlight the limitations of the proposed method and discuss about future improvements/works.

3. The XRD patterns should be refined using the Rietveld method by using relevant software. The relevance factors (chi^2, RB, etc) of XRD patterns fitting should be added. The possible reasons for the variations (increase or decrease) in lattice parameters and crystallites size should be provided in the manuscript.

4. The depth of technical discussions needs to go much further. Vague explanation is provided

5. Authors should compare their results with the recently published articles (2022/2023) in this field.

6. There is total ignorance of magnetic study by the fitting of some equation. You can see the article https://doi.org/10.1002/pssb.201800676 and https://doi.org/10.1007/s00339-019-2615-8

Reviewer #5: PLOS ONE

PONE-D-24-34021

28-8-2024

Solvothermal synthesis of polyvinyl pyrrolidone encapsulated, amine-functionalized

copper ferrite and its use as a Magnetic Resonance Imaging contrast agent

The manuscript reports the preparations of polyvinyl pyrrolidone /amine-functionalized copper ferrite NPs for MR Imaging application.

In general, the manuscript falls well within the scope of Plos One and has its merits to support its consideration of publication from a standpoint of functional materials design and preparations. The following issues please be addressed before the recommendation of publication.

Comments 1.

The abstract could be written with literature values.

Comments 2

The paper failed to explain the novelty of the prepared MRI contrast agent with other reported agents. The author must rewrite the introduction section to properly explain the novelty of synthesis method.

In the Introduction, section, lines 65-66, page 3, There is no citation for the sentence.

lines 65-66, the sentence was not cited.

Comments 3

The properties in terms of physical and biological activity of the fabricated NPs should be compared with other commercially reported contrast agent previously in the main text,. Otherwise, it is difficult to claim the superiority of this product.

Comments 4

Please include the model no, brand name, and origin of instruments used in this work. For example, general information about the Bruker D8 equipment, Bruker Vertex 70 equipment, Weiss-type electromagnet, etc was not given in the paper.

Comments 5

Please follow the same notation in mentioning units like ml, minutes, hours, etc....for example, if authors used hr, hrs, hour, and hours to mention time please check all units and follow a specific notation of short forms for the same.

The author must write properly the subscript in the manuscript. For instance, CuFe2O4-NH2

Comments 6

The influence of cross-linking time and mass ratio of crosslinker on the mechanical performances and also the biological behaviors of the hydrogel can be expanded in a discussion.

Comments 7

In Figure 1, rewrite the caption. It does not clearly explain the figure 1. The abbreviation of PVP and MEA could be included in the material sections. Figure 1 could be named as Scheme 1

I suggest authors to add some molecular indexes analysis to increase the publication value of the paper.

Reviewer #6: Minor revision is required.

English need to be written in standard form. The editing by a native speaker or professionl language editing service must be utilized to correct the overall english writing.

6. PLOS authors have the option to publish the peer review history of their article (what does this mean?). If published, this will include your full peer review and any attached files.

Reviewer #1: **Yes: **Shahrzad Javanshir

Reviewer #2: No

Reviewer #3: No

Reviewer #4: **Yes: **Krutika L Routray

Reviewer #5: No

Reviewer #6: No

---

## [Author Response · Author response to Decision Letter 0]

19 Nov 2024

Reviewer #1: 

While the abstract mentions valuable insights and applications, it could further emphasize the broader implications of this research. How does this advancement contribute to the field of MRI or materials science in general?

1. Grammar and Word Choice:

o Line 29: There are some grammatical errors, such as "improved porperties in case of the of CuFe2O4-NH2 MW particles" which should be corrected to "improved properties in the case of the CuFe2O4-NH2 MW particles." The use of “of” twice in that sentence appears to be a typo.

Thank you for your suggestion, it has been corrected.

o Some sentences could be made more concise. For instance, the phrase "an innovative method was successfully applied and has yielded a novel particle with enhanced properties compared to traditional ferrite-based materials" could be streamlined. Consider rephrasing it to "This innovative method successfully produced nanoparticles with enhanced properties compared to traditional ferrite materials."

2. Materials:

o Please write the chemical formula correctly using subscripts.

Thank you for your suggestion, it has been corrected.

o Ensure you consistently use the same term (e.g., "ethanolamine" vs "ethanol amine") to avoid confusion.

Thank you for your suggestion, it has been corrected.

o Ensure consistency in the naming conventions used for compounds and solvents throughout the document. Example: Polyvinylpyrrolidone (PVP)

Thank you for your suggestion, it has been corrected.

o Line 116: “20 ml of ethanol amine was added” Please convert this to mmol.

It was converted to mmol.

o Line 116: The phrase "continuous continuous mixing" seems to be a typographical error; use "continuous stirring”

Thank you for your suggestion, it has been corrected.

3. Review of Results and Discussion

o The section describes the characterization of amine-functionalized magnetic nanoparticles synthesized through two different methods: reflux and microwave-assisted methods (CuFe2O4-NH2 Refl. and CuFe2O4-NH2 MW). Key aspects of the results are highlighted, including the crystallographic phases identified via X-ray diffraction (XRD) and specific phase compositions derived from Rietveld refinement.

Thank you the suggestion, the follow section was added in the manuscript:

„Summarizing the XRD results, it can be concluded that the nanoparticles produced by microwave synthesis do not contain elemental copper. They contain two magnetizable phases, magnetite and copper ferrite, both of which are well suited as MRI contrast agents. The refluxed sample contains copper particles in addition to the magnetic phases mentioned above. The magnetite has crystallised on the surface of the copper, as copper is a good thermal conductor and a magnetic layer has formed on its surface, this raises the possibility of testing it in magnetic hyperthermia for cancer treatment in the future.”

o The paragraph could benefit from clearer structuring. Consider breaking up long sentences for better readability. For instance, the sentence about the presence of copper peaks could be split to enhance clarity.

The long sentences were broken.

o Use subheadings where appropriate (e.g., “XRD Analysis” or “Particle Size Distribution”) to help readers navigate the section easily.

The proposed subtitles have been added to the manuscript.

o Line 206: "mesarured" which should be "measured,"

Thank you for your suggestion, it has been corrected.

o Provide a brief explanation or background on why the determination of these phases (CuFe2O4, magnetite, Cu) is significant for the application as MRI contrast agents. This context can help readers understand the relevance of the results.

The follows were added XRD results:

“Using two different synthesis methods, CuFe2O4-NH2 Refl. and CuFe2O4-NH2 MW were prepared, these methods may result in differences in phase compositions. In order to confirmation, that the sample consisted of nanoparticles with the magnetic properties we wanted to produce, we needed to perform XRD measurements. XRD helps to detect possible contaminants (non-magnetic components) that may degrade the quality of the contrast agent, such as hematite or copper-oxides.”

4. Ensure all statements regarding background information and previous research are properly cited.

5. Minor grammatical errors and sentence structure issues could be addressed for a more polished presentation. The overall tone is professional, but a thorough proofreading would enhance clarity. 

Thank you, we have made the linguistic corrections to the manuscript, and the English of the manuscript has been much improved.

6. Incorporating an MTT assay in your study of copper ferrite nanoparticles would be a strong choice to assess the cytotoxicity, provided it is part of a broader strategy that may include additional assays for validation.

AlmarBlue assay was assessed. The purpose of this was to show if Copper-ferrite would be an optimal platform for further development: 1) as a targeted therapeutic agent, 2) capable to provide contrast. We tested the toxicity on HEK293 cells for 24 hours, where we could detect a decreased cell number as the results of incorporating Copper atoms into the nanoparticle sample (CuFe2O4-NH2 Refl). These results are included in the supplemental material of this manuscript. However, 

After a bolus injection into the tail vein of mice, a rapid dilution of twenty-times takes place. In vivo, acute effects or toxic effect were not observed after the injecting the particles into mice. 

A more detailed toxicological study will be published on Copper-, Cobalt-, Zinc-ferrite and other particles, discussing their short- and long-term effects as well. The magnetite-coated copper nanoparticles will be tested later in hyperthermia cancer treatment in mice. This will also include toxicological tests, but these have not yet been carried out.

7. A comparison with other published work must be performed to contextualize the findings and assess their significance concerning existing literature on similar nanoparticles." Especially with your published article https://doi.org/10.3390/ijms242216203 what is the impact of Cu compared to Zn

Thanks for the suggestion, the results of zinc ferrite synthesis were compared with the results of copper ferrite synthesis. We have also explained why no pure spinel phase was obtained in the production of copper ferrite.

“In a previous publication, we reported on the preparation of zinc ferrite and its application as an MRI contrast agent. The synthesis methods were very similar to those used in the present work for the preparation of copper ferrite. In contrast to the present results, no other phases were found in the sample for zinc ferrite, only the spinel was formed during both preparation methods [https://doi.org/10.3390/ijms242216203]. In the case of copper(II) ions, reduction can occur, as ethylene glycol is an effective reducing agent in the conversion of Cu(II) ions to elemental copper nanoparticles. Ramyadewi et al. refluxed Cu(II) salt in ethylene glycol solution at 200 °C (as we did) and produced copper nanoparticles [https://doi.org/10.1016/j.matlet.2011.12.055].”

8. The conclusions section should give the importance and usefulness of the research results

We added the concluding part with the consideration that, our experiments have shown that the refluxing process can be used to successfully produce particles with a core-shell structure, with copper particles inside (which are good thermal conductors) and nanoparticles on the outside. This result raises the possibility that these core-shell particles could be used as magnetic hyperthermia treatment for cancers.

Reviewer #2: 

1. Add miller indices in xrd patterns

Thank you, the suggestion, the Miller indices were added to the XRD patterns.

2. Add error bar on statistical curve

Error bars were added to the corresponding MRI figure (Fig. 11)

3. Compare each obtained data with similar work

The obtained results (XRD and VSM measurements) were compared with other works.

4. The reason of using each precursor should be mentioned in exp. Section

“In the course of our previous work, we have produced magnetic nanoparticles many times and found that ferrites can be efficiently produced starting from nitrate salts [https://doi.org/10.1038/s41598-024-54792-5] [https://doi.org/10.3390/ijms241713347].”

5. Conclusion should be written more comprehensive

Reviewer #3: 

The paper „Solvothermal synthesis of polyvinyl pyrrolidone encapsulated, amine-functionalized copper ferrite and its use as a Magnetic Resonance Imaging contrast agent“, written by Fatemeh Heydari et al., presents copper ferrite nanoparticles for contrast-enhanced MRI and a novel method of their preparation.

The authors present an improved synthesis method and compare nanoparticles prepared by a standard solvothermal synthesis and modified, microwave-assisted procedure. Particles prepared by both ways were compared by a row of physical measurements (magnetometry, X-ray diffraction, dynamic light scattering, relaxometry), and tested contrast enhanced MRI in vitro and in vivo. However, it is not clear, what is the main message: new particles for MR imaging, or the new procedure for preparation? Incorporation of copper into the nanoparticles might be interesting, but not from the point of view of MR imaging. Standard commercially available nanoparticles based on iron oxides have better performance (same or higher r2 relaxivity, substantially higher r1 relaxivity). If there is no special function of the copper in the nanoparticles (or of the particles themselves), there is no reason to use them instead of standard iron oxide nanoparticles. Therefore, I suppose, that the authors should emphasize the positive point of their work – the novel preparation method, thoroughly discuss the physical properties of nanoparticles prepared in a new way, discuss possible use of the method for preparation other nanoparticle types, and possibly think of submitting the paper to more technically oriented journal. I suppose that it should be rather rewritten than revised. I am sorry.

Specifically:

I appreciate a row of physical methods for characterization of nanoparticles; nevertheless, the paper suffers of many weak parts:

Language: I am not a native speaker, but I dare to say that the manuscript requires a thorough proofread. Many typos, mistakes or strange sentences strike even me. There are also typographic errors (physical quantities should be in italics, the authors should use subscript for numbers in chemical formulas, superscript for exponents etc.) throughout the text. Also, I think that there are wrong references to the images in the text

Point by point comments:

1) p. 3, l. 29: There should be plural of the verb “show”, I suppose. Typo in “properties”

2) p. 3, l. 30-32: Word order is strange.

The word order has been changed, the sentence was rephrased.

3) p. 4, l. 77-81: I have to remind, that NFS was extremely rare and since identifying the cause of NSF and adoption of relevant measures (such as avoiding application of linear Gd chelates to patients with renal failure), no case of NFS related to Gd contrast agents was reported. On the other hand, iron-oxide nanoparticles have much more frequent side effects (albeit not fatal) and there is no nanoparticle-based contrast agent currently approved for clinical use in Europe. Therefore, dreams of replacing Gd-based contrast agents by magnetic nanoparticles in near future are just wishful thinking (I do not oppose to study and explore magnetic nanoparticles, they are for sure very interesting and useful, but, please, do not defame Gd contrast agents, which are currently irreplaceable and I believe that they will be replaced by novel native MR sequences rather than by nanoparticles).

Despite the reviewer’s suggestions, we still believe that other contrast materials besides Gd-chelates will be used in clinical practice in the near future. Of course, clinical trials and thorough examinations of such materials take time, but it is achievable. Economic aspects and significant advantages compared to Gd-materials must also be present. Additionally, the project should be backed by stakeholders and pharmaceutical companies. Our main goal is to describe these materials as thoroughly as possible, highlighting the potential pipeline towards the optimal candidate(s) for the future. Without wishful thinking, no change is achievable. Thank you very much for the comment.

4) p. 4, l. 95-96: “Capping” agents or molecules is quite strange, I would recommend “coating”

5) p. 5, l. 117: Please, use RCF instead of RPM

6) p. 6, l. 133: “Techniques”, not “technics”

Thank you for your comments, grammatical errors have been corrected in the manuscript based on the reviewers' suggestions.

7) p. 8, l. 189-190: In vivo experiments are insufficiently described. How many mice the authors used? Which strain? Age? Sex? How was the suspension applied? Volume? Concentration? (Part of this information is in the Result section, but it should be here!)

We have included the relevant information to the “Characterization Techniques” section. 

“For the in vivo measurements, n=2 female, 10-week-old BalbC mice were used. The 0.2 mL of the 1 mg/mL contrast material was injected into the tail vein of the animals.”

8) p. 8, l. 190-193: The described sequence (with a 90° excitation pulse and TR = 270 ms) has also quite strong T1-weighting, I suppose. Please, comment whether it may affect the results. Which organs were examined? How was the signal calculated/compared?

The reason why the in vivo scans have a slight T1-weighting is that the sequence parameters were choosen to have fairly high liver signal on the native pre scans. That’s how the signal decrease could have been sufficiently high. As the r1 relaxivities are very low, we didn’t expect to have signal increase due to the T1 shortening effect even with the used paratmeter set. We have also added the following description to the manuscript: 

“In InterView Fusion software (Mediso Ltd) 8 ROIs (liver, spleen, visceral fat tissue, renal pelvis, brain, muscle in the leg, cortex and medulla of the kidneys) were manually drawn on the pre scans of each mice. Images were normed to have same signal intensities in the muscle in order to have comperable signal intensity (SI) values. Relative signal changes were calculated in each ROI by the formula (SIpre – SIpost) / SIpre to determine the biodistribution of the nanoparticles. The higher the measured signal intensity decrease the more the accumulated nanoparticles there.”

9) p. 9: Figure 1 has no caption. Nevertheless, I think this figure is redundant.

The function of figure one is to illustrate and summarize the whole study. We have corrected the mentioned figure and modified its caption. We added an explanation in the text. It has already become useful because it clearly illustrates the partial processes of our experimental work.

10) p. 9, l. 214: The caption to Fig. 2 refers to A and B figures, however, there is only one graph in Fig. 2.

Thank you for your comment, incomplete figure has been replaced.

11) p. 10, Table 1: How many crystallites was evaluated to obtain average size (should be described in Methods)

Two methods were used to determine the particle sizes, based on XRD results and TEM images. These results were compared.

The crystallite sizes in Table 10 were determined from XRD measurements, the method is described in the section Characterization techniques: 

“Average crystallite size of the domains was calculated by the mean column length calibrated method using the full width at half maximum (FWHM) and the width of the Lorentzian component of the fitted profiles.” 

The following addition has been made to the manuscript to clarify the measurement method based on the TEM recordings:

„For the size analysis based on the TEM images, 100 particles were measured using a scalebar of pixel ratios from the TEM images. For the manual size determination, we used Image J [ImageJ: Image processing and analysis in Java WS Rasband - Astrophysics Source Code Library, 2012 - ui.adsabs.harvard.edu] [DOI: 10.17577/IJERTV4IS110211] https://doi.org/10.1016/j.compag.2008.02.007.”

12) p. 10-11: What is the difference between Fig. 3 and 4? The caption is almost th

---

## [Decision Letter · Decision Letter 1]

9 Dec 2024

Solvothermal synthesis of polyvinyl pyrrolidone encapsulated, amine-functionalized copper ferrite and its use as a Magnetic Resonance Imaging contrast agent

PONE-D-24-34021R1

Dear Dr. Forgách,

We’re pleased to inform you that your manuscript has been judged scientifically suitable for publication and will be formally accepted for publication once it meets all outstanding technical requirements.

Kind regards,

Amir Elzwawy, Ph.D.

Academic Editor

PLOS ONE

Additional Editor Comments (optional):

The manuscript is acceptable now as the reviewers recommended, consider any further needed points regarding ethical concerns or technical ones, and follow the journal guidelines for the completion of your article production. Congratulations.

Reviewers' comments:

Reviewer's Responses to Questions

**Comments to the Author**

1. If the authors have adequately addressed your comments raised in a previous round of review and you feel that this manuscript is now acceptable for publication, you may indicate that here to bypass the “Comments to the Author” section, enter your conflict of interest statement in the “Confidential to Editor” section, and submit your "Accept" recommendation.

Reviewer #2: All comments have been addressed

Reviewer #4: All comments have been addressed

2. Is the manuscript technically sound, and do the data support the conclusions?

Reviewer #2: Yes

Reviewer #4: Yes

3. Has the statistical analysis been performed appropriately and rigorously? 

Reviewer #2: Yes

Reviewer #4: N/A

4. Have the authors made all data underlying the findings in their manuscript fully available?

Reviewer #2: Yes

Reviewer #4: Yes

5. Is the manuscript presented in an intelligible fashion and written in standard English?

Reviewer #2: Yes

Reviewer #4: Yes

6. Review Comments to the Author

Reviewer #2: All comments I sent before are considered in the revised version.

Finally, in my opinion, this is good work to be published and I recommend publication of this paper in your journal as it is

Reviewer #4: (No Response)

7. PLOS authors have the option to publish the peer review history of their article (what does this mean?). If published, this will include your full peer review and any attached files.

Reviewer #2: No

Reviewer #4: **Yes: **Dr. Krutika L Routray

---

## [Editor Report · Acceptance letter]

16 Dec 2024

PONE-D-24-34021R1 

PLOS ONE

Dear Dr. Forgách, 

I'm pleased to inform you that your manuscript has been deemed suitable for publication in PLOS ONE. Congratulations! Your manuscript is now being handed over to our production team.

Kind regards, 

on behalf of

Dr Amir Elzwawy 

Academic Editor

PLOS ONE